# Ephrin-A/EphA specific co-adaptation as a novel mechanism in topographic axon guidance

**Felix Fiederling, Markus Weschenfelder, Martin Fritz, Anne von Philipsborn[†], Martin Bastmeyer, Franco Weth\***

Department of Cell and Neurobiology, Karlsruhe Institute of Technology, Zoological Institute, Karlruhe, Germany

**\*For correspondence:** franco. weth@kit.edu

**Present address:** [†]Danish Research Institute of Translational Neuroscience, Aarhus, Danmark

**Competing interests:** The authors declare that no competing interests exist.

**Abstract** Genetic hardwiring during brain development provides computational architectures for innate neuronal processing. Thus, the paradigmatic chick retinotectal projection, due to its neighborhood preserving, topographic organization, establishes millions of parallel channels for incremental visual field analysis. Retinal axons receive targeting information from quantitative guidance cue gradients. Surprisingly, novel adaptation assays demonstrate that retinal growth cones robustly adapt towards ephrin-A/EphA forward and reverse signals, which provide the major mapping cues. Computational modeling suggests that topographic accuracy and adaptability, though seemingly incompatible, could be reconciled by a novel mechanism of coupled adaptation of signaling channels. Experimentally, we find such 'co-adaptation' in retinal growth cones specifically for ephrin-A/EphA signaling. Co-adaptation involves trafficking of unliganded sensors between the surface membrane and recycling endosomes, and is presumably triggered by changes in the lipid composition of membrane microdomains. We propose that co-adaptive desensitization eventually relies on guidance sensor translocation into *cis*-signaling endosomes to outbalance repulsive *trans*-signaling.

## Introduction

The dazzling diversity of functions of the nervous system are brought about by neural networks of definite connectivity, most of which arise during ontogenesis through targeted outgrowth of axons. This is accomplished by growth cones (GCs) at the tips of growing axons (*Lowery and Van Vactor, 2009*; *Vitriol and Zheng, 2012*), which sense genetically encoded chemotactic guidance cues (*Dickson, 2002*; *Kolodkin and Tessier-Lavigne, 2011*). Such genetic hardwiring endows the individual at prefunctional stages with computational architectures, which have been fitted for survival through the evolution of the whole species.

A well-studied example is the development of the retinotectal projection (*Figure 1*), which, in non-mammalian vertebrates, connects the retinal ganglion cells (RGCs) of the eye with the midbrain's optic tectum in a topographic, *i.e.*, neighborhood-preserving manner (*Feldheim and O'Leary, 2010*; *Lemke and Reber, 2005*; *Weth et al., 2014*). Research on this two-dimensional mapping has mainly focused on the anterior-posterior axis, whereby the temporal retina is projected onto the anterior tectum and the nasal retina onto the posterior tectum. The major guidance cues along this axis are EphA receptor tyrosine kinases and their glycosylphosphatidylinositol-(GPI)-anchored ephrin-A ligands (*Suetterlin et al., 2012*; *Triplett and Feldheim, 2012*). EphAs and ephrin-As are counter-graded on both the retina and the tectum (temporal retina — EphA high, ephrin-A low; anterior tectum — EphA high, ephrin-A low [*McLaughlin and O'Leary, 2005*]). The ephrin-A/EphA system can signal in forward (EphA acting as receptor) and in reverse (ephrin-A as

**eLife digest** The human brain contains roughly 100 billion neurons, which are organized into complex networks. But how does the brain establish these networks in the first place? Neurons have long projections known as axons and, in the developing brain, these axons form structures called growth cones at their tips. The growth cones possess finger-like appendages that probe their surroundings in search of signals displayed on the surface of other cells. These signals guide the growth cones to their targets and move the axon tip into a position where it can form connections with other neurons within a particular network.

The signals that growth cones follow are often distributed in concentration gradients so that the levels of a signal may be low at one end of a brain structure and gradually increase to a maximum level at the other end. In the developing visual system, for example, about one million axons from the retina reach their proper targets in visual regions of the brain by reading gradients of signals called ephrins and Ephs. However, when Fiederling et al. studied retinal neurons in a petri dish, they found that the axons became much less sensitive to both signals upon prolonged exposure to them. This unexpected finding raised a new question. If neurons rely upon these gradients for navigation, how do they continue to find their way if they also become less sensitive to those signals over time?

Fiederling et al. used a computer to simulate the events occurring in the developing brain. The simulations were based on the idea that navigating growth cones sense the ratio of ephrins to Ephs, instead of sensing the individual concentrations of these signals. Thus, by keeping the amounts of all involved sensors in strict proportion to each other while continuously re-adjusting them, the axons could still be accurately guided to their targets even though the neurons would become less sensitive to the signals. Experiments in neurons grown in petri dishes confirmed that retinal growth cones do exactly this and regulate the amounts of ephrin and Eph sensors on their outer membranes in a highly coordinated manner using a previously unknown mechanism.

Given that signaling requires energy, the brain may have evolved this system to reduce the costs associated with wiring itself up. The system also offers greater flexibility than guidance based on the absolute concentrations of the signals. If other regions of the brain use a similar mechanism to establish their own wiring patterns, then understanding such basic mechanisms might eventually provide insights into diseases of miswiring such as schizophrenia and autism.

receptor) directions (*Egea and Klein, 2007*; *Kania and Klein, 2016*; *Lisabeth et al., 2013*). Both signaling channels act repulsively on RGC GCs. Receptor/ligand interactions typically occur in *trans*, between different cells, but, when present on the same cell, as on RGCs, they can additionally happen in *cis* (*Hornberger et al., 1999*). We have recently proposed a simple but powerful comprehensive computational model (*Gebhardt et al., 2012*) that includes all possible ephrin-A/EphA interactions in this system (fiber–target, fiber–fiber, *cis*, each forward and reverse). The model assumes that axon targeting aims to balance all summed reverse signals against all summed forward signals sensed by the GC. There have been diverse further attempts to explain retinotopic mapping (*Hjorth et al., 2015*; *Simpson et al., 2009*). Most of them agree, however, that topographic guidance is based on quantitative signaling, whereby the concentrations of sensors and cues bear the topographic information. This assumption has also been corroborated by experimental evidence (*Baier and Bonhoeffer, 1992*; *Hansen et al., 2004*; *Rosentreter et al., 1998*; *von Philipsborn et al., 2006b*).

On the other hand, it has been shown repeatedly that GCs can adapt to chemotactic guidance cues. *Xenopus* spinal GCs, in turning assays, can adapt to attraction by soluble Netrin-1 or brain-derived neurotrophic factor (BDNF) (*Ming et al., 2002*). Ligand-specific desensitization was correlated with decreased $Ca^{2+}$ signaling, whereas re-sensitization required MAP-kinase activation and local protein synthesis. *Xenopus* retinal axons in collapse assays were shown to adapt towards the collapse-inducing activities of soluble Sema3A and Netrin-1 (*Piper et al., 2005*). Here, ligand-specific desensitization and re-sensitization, depended on endocytosis and protein translation, respectively. Notably, however, none of these guidance cues has been involved in topographic mapping. Indeed, it is difficult to imagine how adaptive signal modulation might be compatible with the need

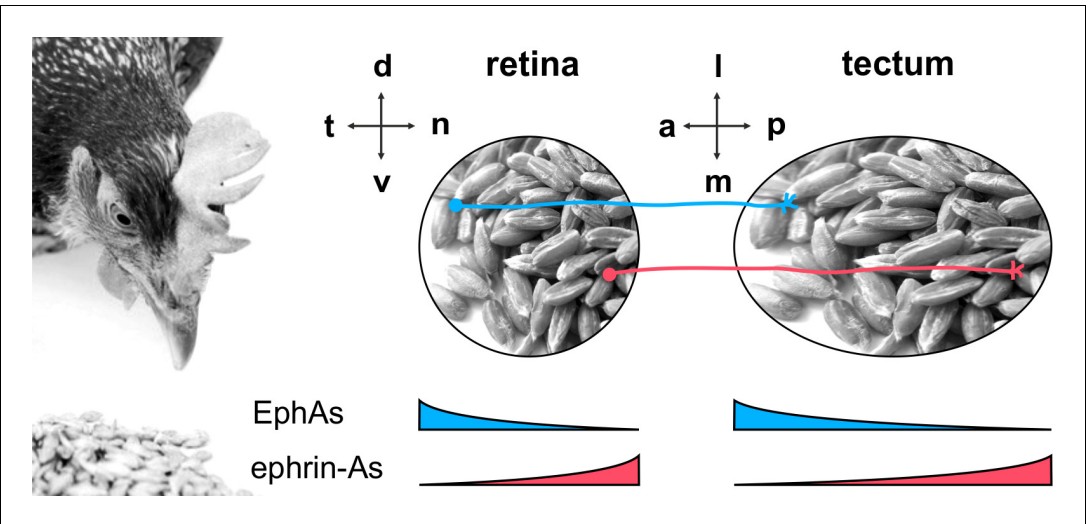

**Figure 1.** The retinotectal projection. The topographic projection in the chicken visual system connects RGCs from the retina to the midbrain's optic tectum. The temporal/nasal (t/n) axis of the retina is mapped onto the anterior/posterior (a/p) axis of the tectum, whereas the retinal dorsal/ventral axis projects onto the lateral/medial axis of the tectum. Retinal GCs are guided to their tectal targets by repulsive signals from counter-gradient distributions of tectal ephrin-As (red; p>a) and EphAs (blue; a>p). Detection of these cues is mediated via retinal EphA and ephrin-A sensors, which are expressed in t>n and n>t gradients, respectively.

for quantitative signaling. First evidence, however, indicates that retinal GCs might in fact be able to adapt to topographic cues (*Rosentreter et al., 1998*; *von Philipsborn et al., 2006b*). We therefore decided to investigate this conundrum in more detail.

Introducing a novel adaptation assay ('gap assay'), we demonstrate that chick retinal GCs adapt towards both ephrin-A forward and EphA reverse topographic signals. Led by computational modeling, we postulate a distinctive novel type of signal modulation ('co-adaptation') to reconcile adaptation and topographic accuracy. We prove experimentally that such ephrin-A/EphA specific co-adaptation is indeed realized in retinal GCs. Using SNAP-tagged ephrin-A5 to label sensor surface populations, pharmacological inhibition, and co-localization studies with Rab11, we show that co-adaptation on the cellular level involves trafficking of the guidance sensors between the GC membrane and the recycling endosome. By inducing adaptation by altering the lipid composition of the GC membrane, we provide evidence that repartitioning of guidance sensors between membrane microdomains might drive co-adaptation downstream of ephrin-A/EphA signaling. Eventually we propose, on the basis of our computational model, a hypothetical mechanism of co-adaptation, suggesting that an increase in the efficiency of *cis*-signaling from endosomes eventually desensitizes GCs against *trans*-signaling, without altering their topographic identity.

## Results

### Retinal GCs adapt towards topographic EphA forward and ephrin-A reverse signals

In *in-vitro* collapse assays (*Cox et al., 1990*; *Kapfhammer et al., 2007*) typically ~80% of temporal GCs show a collapsed morphology after 20 min incubation with 0.25 µg/ml ephrin-A5-Fc, whereas control (Fc fragment)-treated GCs mostly remain intact (*Figure 2A,B*). Surprisingly, however, we observe that temporal GCs recover their morphology, despite the presence of the repulsive cue, after prolonged incubation. After 120 min of incubation with ephrin-A5-Fc, the fraction of collapsed GCs drops from 77.1% (20min) to 32.9% and, thus, to control levels (20 min Fc: 23.3%; 120 min: 27.2%; *Figure 2A*). This recovery is not caused by a loss of activity of the repulsive cue, as a medium containing 0.25 µg/ml ephrin-A5-Fc that has been on a first culture for 120 min triggers full response

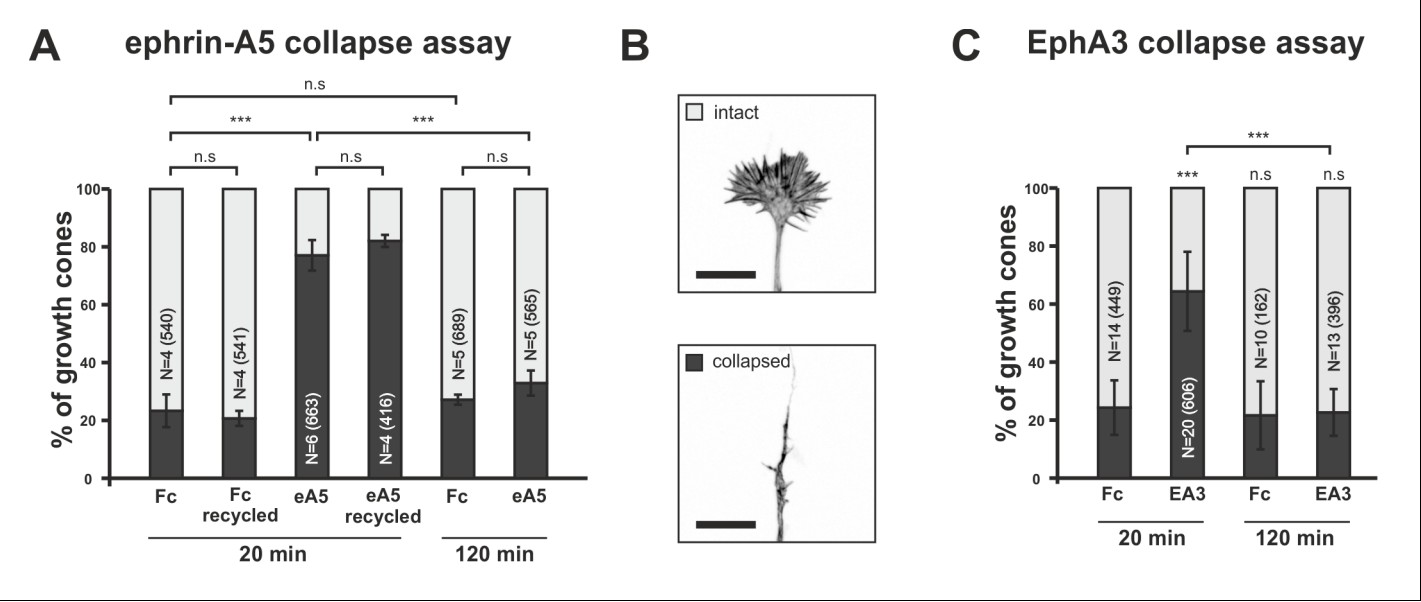

**Figure 2.** Growth cone desensitization towards soluble ephrin-A5 and EphA3. (**A**) Temporal GCs initially collapse upon application of 250 ng/ml ephrin-A5-Fc (eA5 20 min: 77.1%), but recover their morphology within 120 min in the presence of the cue (eA5 120 min: 32.9% collapsed GCs). Ephrin-A5-Fc is still functional after incubation for 120 min, when re-used on a fresh culture (eA5 recycled 20 min: 81.5% collapsed). Fc controls demonstrate the specificity of the effect of ephrin-A5 (Fc 20 min: 23.3%; Fc recycled 20 min: 20.7%; Fc 120 min: 27.2%). (**B**) Representative intact and collapsed GC morphology (Phalloidin Alexa488). Scale bar: 10 µm. (**C**) Soluble EphA3-Fc triggers a collapse when administered at 15 µg/ml (EA3 20 min: 64.4% collapsed), whereas an equimolar concentration of Fc (4.2 µg/ml) does not (Fc 20 min: 24.3%). GCs desensitize towards EphA3-Fc within 120 min (EA3 120 min: 22.6%). Combined data from nasal and temporal GCs. N: number of independent experiments; number of analyzed GCs in brackets. Error bars represent standard deviations. T-test with n.s.: $\alpha \geq 0.05$, ***: $\alpha < 0.001$.

The following source data and figure supplements are available for figure 2:

**Source data 1.** Original data underlying bar charts of *Figure 2A, C*.

**Figure supplement 1.** EphA3 collapse assays and ephrin-A5/EphA3 dissociation constants.

**Figure supplement 1—source data 1.** Original data underlying bar charts of *Figure 2—figure supplement 1*.

when reused on a second culture for 20 min (81.5% collapsed temporal GCs; *Figure 2A*). Therefore, recovery in the presence of ephrin-A indicates desensitization of RGC GCs towards forward signals.

To investigate reverse signaling desensitization, we first established EphA3 collapse assays. At high concentrations (15 µg/ml), a specific and significant collapse is triggered (64.4% of GCs; equimolar Fc: 24.3% collapsed; *Figure 2C*), but this does not occur at lower concentrations (*Figure 2— figure supplement 1A*). We suggest that the exclusive response to high EphA concentrations is due to an exclusive activity of EphA3 oligomers (n > 2), which might form in concentrated solutions only. Supporting evidence comes from measuring ephrin-A5/EphA3 affinity by biolayer interferometry. Remarkably, binding constants depend on the EphA3 concentration and are dramatically increased at high concentrations (≥15 µg/ml; *Figure 2—figure supplement 1B*). The assumed oligomers remain stable in solution for at least 1 day (*Figure 2—figure supplement 1C*). Notably, but similar to the response to ephrin-A5, GCs completely recover after prolonged incubation with EphA3-Fc (22.6% collapsed after 120 min; Fc controls after 120 min: 21.6%; *Figure 2C*), demonstrating GC desensitization also towards EphA3 reverse signals.

Full adaptability comprises desensitization and re-sensitization. To address the latter and to better approximate the *in-vivo* situation regarding surface-bound cues, we developed a novel *in-vitro* adaptation assay, the 'gap assay'. To this end, we manufactured patterned guidance substrates by protein contact printing (*von Philipsborn et al., 2006a*), which consisted of two rectangular fields homogeneously covered with the cue and separated by a permissive gap (laminin) of defined width.

To ensure the activity of the printed cue, positive controls are performed in which axons grow from homogeneous laminin towards an identical field of the cue. For ephrin-A5-Fc, naïve temporal axons show a robust stop reaction at the boundary to the ephrin field in such controls (98% stopping; *Figure 3A*). Negative controls using a corresponding field of human Fc fragment (28.8% stopping) demonstrate the specificity of the response (*Figure 3C*). By contrast, axons starting from a field of homogeneous ephrin-A5-Fc no longer stop and continue growing onto the second ephrin field if the gap is small (*Figure 3A'*), revealing their desensitization in agreement with the collapse assays. Notably, however, the proportion of GCs that stop in front of the second ephrin field increases with increasing gap sizes. Only 15.3% of temporal axons stop in assays with 50 µm wide gaps, but 43.0% stop after 75 µm, 59.2% after 100 µm and 81.0% after 200 µm wide gaps (*Figure 3A'' and C*), indicating that GCs regain sensitivity towards forward signaling while growing on the permissive gap.

In reverse signaling gap assays, naïve nasal RGC GCs show a clear stop reaction at the boundary to a field of contact printed EphA3 (92.6% stopping; *Figure 3B*). GCs no longer stop in front of an identical field after a small gap when starting on EphA3 (65 µm: 25.0% stopping; *Figure 3B'*), indicating desensitization towards reverse signals in accordance with the collapse assays. As in the forward signaling gap assays, they gradually regain sensitivity with increasing gap sizes (90 µm: 39.9% stopping; 115 µm: 60.1% stopping; 215 µm: 75.8% stopping; *Figure 3B'' and C*), demonstrating re-sensitization towards reverse signaling. Re-sensitization of forward and reverse signaling in gap assays displays very similar dependencies on the overgrown distance hinting at a potential common mechanism.

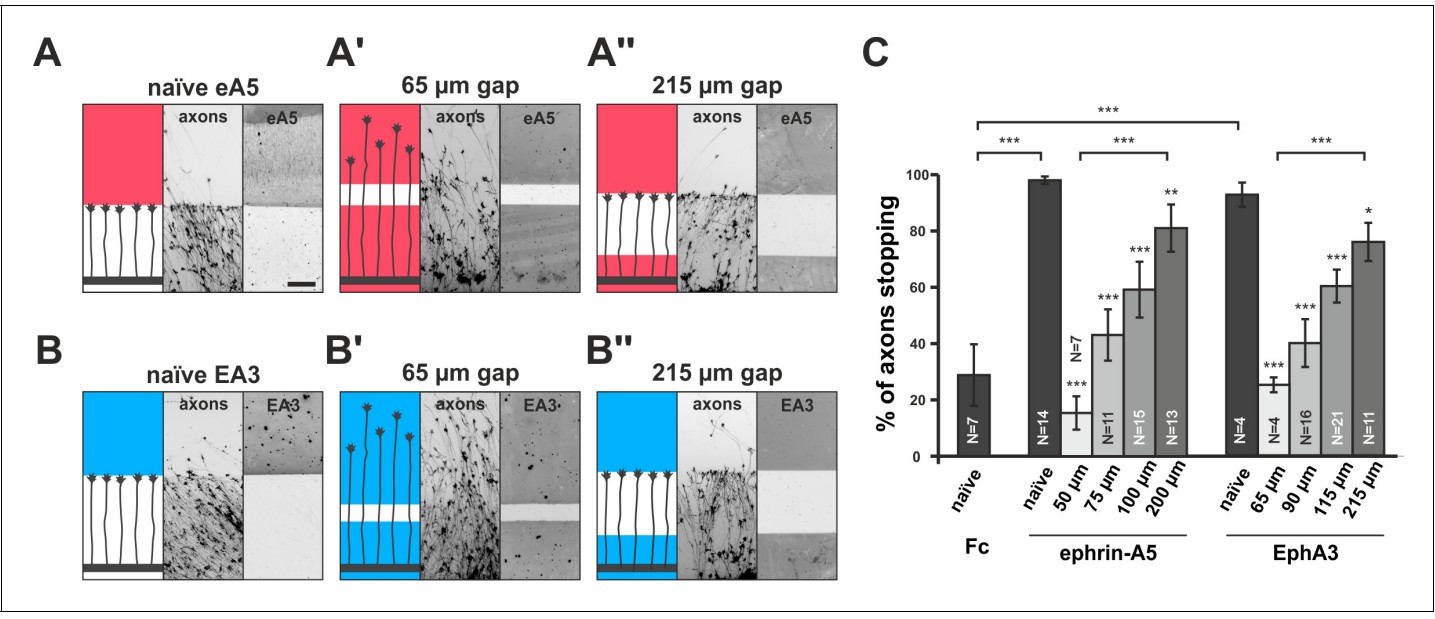

**Figure 3.** Growth cone adaptation towards substrate-bound ephrin-A5 and EphA3. Subfigures in (**A**) and (**B**) each display a cartoon (left) illustrating the experimental setup consisting of explant (thick black strip), axons and printed guidance protein (colored field(s)), the inverted signal of fluorescent phalloidin-stained axonal actin (middle; explant not shown) and the underlying, antibody-labeled substrate (right) are shown in a detailed view of a representative microscopic image (scale: 100 µm). (**A**) Naïve temporal axons stop in front of a homogeneous field of ephrin-A5-Fc (eA5 15 µg/ml; red), but do not react to an identical boundary when initially grown on ephrin-A5 after a 65 µm wide gap (**A'**). Stopping behavior returns in axons having crossed a 215 µm wide, ephrin-free gap (**A''**). (**B–B''**) Nasal RGC axons show very similar behavior on EphA3-Fc (EA3 15 µg/ml, blue) gap substrates as described for ephrin-A5 gap assays. (**C**) Quantification of gap assays. Stop reactions were quantified as the percentage of fibers not entering the protein field after the gap. Fc naïve: 28.8% stopping. eA5 naïve: 98.0%, eA5 50 µm: 15.3%, eA5 75 µm: 43.0%, eA5 100 µm: 59.2%, eA5 200 µm: 81.0% stopping. EA3 naïve: 92.6%, EA3 65 µm: 25.0%, EA3 90 µm: 39.9%, EA3 115 µm: 60.1%, EA3 215 µm: 75.8% stopping. Ephrin-A5 gap assay patterns differ slightly from EphA3 patterns in gap width as a result of a modified stamp geometry. N: number of independent experiments. Error bars represent standard deviations. T-test with *: $\alpha$ <0.05, **: $\alpha$ <0.01, ***: $\alpha$ <0.001.
The following source data is available for figure 3:

**Source data 1.** Original data underlying bar chart of *Figure 3C*.

Together, our results show that topographically mapping retinal GCs robustly adapt towards forward and reverse ephrin-A/EphA signaling. This is puzzling, because topographic guidance is believed to rely on precise quantitative sensing.

## Modeling suggests co-adaptation as a novel mechanism for reconciling growth cone adaptation and topographic accuracy

To conceive how adaptation and topography could be reconciled, we made use of our previous computational model (*Gebhardt et al., 2012*). Briefly, the model (*Figure 4A*) is exclusively based on ephrin/Eph signaling and assumes that GCs do not distinguish the diverse sources of the impinging signals (target cells, other fibers, GC's own surface). They just sense *total* reverse and *total* forward signaling. GC targeting is then considered as a potential minimization process, comparable to a sphere rolling into a well. To describe this process, a 'guidance potential', $D$, is defined, which is conceived to be proportional to the *ratio* of total reverse to total forward signaling and to be minimized when both are equal, indicating target arrival. The effective strengths of the signals received by an individual GC depend on its endowment with sensors (ephrins for reverse and Ephs for forward signaling). Thus, the GC's own ephrin/Eph ratio corresponds to its topographic imprint, determining where on the target area reverse/forward signaling balance will be reached. Therefore, to retain topography in the face of adaptation, which is expected to modulate signal strength, this ratio must remain basically unaltered. In the model, this is achieved by introducing the same multiplicative adaptation factor, $a$, simultaneously to *both* reverse and forward signaling. For more mathematical detail see 'Material and methods'.

After implementing this form of concomitant adaptation of two signaling channels, which we term co-adaptation, the model is still able to form an accurate topographic map (*Figure 4B*, gray circles). By contrast, uncoupled adaptation by independent negative feedback to each channel completely abolishes map formation (*Figure 4B*, white circles). The updated model reproduces the experimental results of the gap assays (*Figure 4C*). This is achieved without changing the model's global explanatory power, described earlier (*Figure 4—figure supplement 1*; *Gebhardt et al., 2012*; *Weth et al., 2014*).

The co-adaptation concept predicts that each channel should be modulated upon its own activation, but counter-intuitively also upon activation of the parallel channel even in the absence of its own ligand. Thus, ephrin-adapted axons should ignore Eph in a 'double-cue' gap assay with small gap size. Similarly, Eph-adapted axons should ignore ephrin (simulations in *Figure 4D*).

## EphA forward signaling actually co-adapts ephrin-A reverse signaling and *vice versa*

To test this prediction, we used double-cue gap assays with substrates comprising different cues on either side of the gap. In ephrin-A5/EphA3 double-cue gap assays, ephrin-A5-adapted temporal GCs, after a small gap (<100 μm), ignore the EphA3 field (26.9% stopping), in front of which they naïvely stop (93.4% stopping; *Figure 5A,A' and D*). As these GCs have not experienced substrate-derived reverse signals before, their desensitization towards these signals can only be explained through co-regulation of reverse and forward signaling upon activation of the forward channel.

Conversely, in EphA3/ephrin-A5 double-cue gap assays (<100 μm), nasal GCs co-adapt their forward signal by growing on an EphA3 reverse signaling field (40.3% stopping; naïve: 91.6%; *Figure 5B,B' and D*). Consistent with the findings in single-cue gap assays, the degree of co-adaptation abates with increasing gap width (eA5–EA3 >100μm: 76.3% stopping; EA3–eA5 >100μm: 77.1% stopping; *Figure 5D*). Co-adaptation is ephrin/Eph-specific, as for example, ephrin-A5-adapted temporal GCs remain sensitive to other repulsive guidance molecules such as Sema3A (naïve: 93.7% stopping, eA5–S3A <100μm: 81.9% stopping; *Figure 5C,C' and D*). Although retinal GCs can efficiently desensitize towards Sema3A in gap-assays (*Figure 5—figure supplement 1A–D*), Sema3A-adapted GCs remain sensitive to ephrin-A5 (*Figure 5—figure supplement 1E*), further attesting to the orthogonality of these signaling pathways and to the specificity of the co-adaptation mechanism for the ephrin-A/EphA-based topographic mapping system.

Together, these findings, being in full agreement with the predictions of our computational model, prove the existence of a novel mechanism of signal modulation (co-adaptation), which allows for topographic mapping in the presence of CG adaptation.

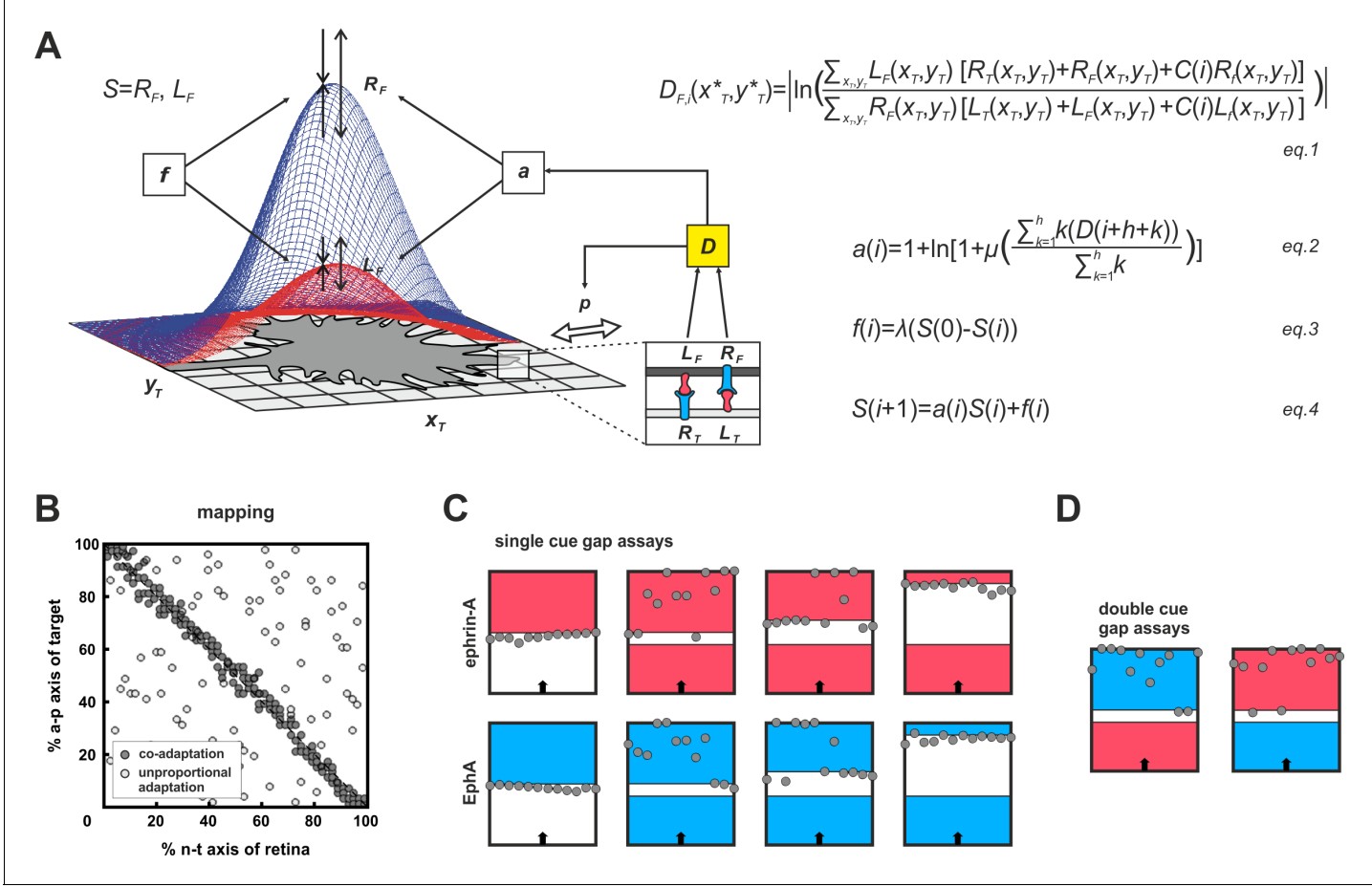

**Figure 4.** Modeling growth cone adaptation and topographic mapping. (**A**) Fiber terminals are modeled as circular discs bearing Gaussian-shaped distributions of EphAs ($R_F$, blue) and ephrin-As ($L_F$, red), according to their retinal origin, moving on a rectangular target field of unit squares ($x_T$, $y_T$), carrying target-derived guidance cues ($L_T$ and $R_T$). Fiber–target and fiber–fiber (*cis* and *trans*) interactions and the resulting forward and reverse signals integrate into a guidance potential, $D$ (**equation1**), which determines the probability, $p$, of a change in position (only fiber–target interactions are illustrated in the inset). Additionally, $D$ is used to calculate an adaptation coefficient, $a$ (**equation2**), which simultaneously modulates $R_F$ and $L_F$ (collectively called $S$; **equation4**) through 'co-adaptation'. A resetting force, $f$ (**equation3**), counteracts $a$. For mathematical detail see 'Material and methods'. (**B**) Mapping plots of simulations with 200 fiber terminals using different implementations of adaptation. The anterior-posterior (a-p) position of a terminal on the target is plotted as a function of naso-temporal (n-t) origin (perfect topography is indicated by all terminals targeting the main diagonal). Terminals that are enabled to regulate sensors independently from each other by canonical adaptation are widely scattered across the target field (white circles), whereas co-adapted terminals (regulating both sensors concomitantly) find their topographically correct target positions (gray circles). In the basic model, the potential minimum is at a position where fiber and target ephrin/Eph concentrations match. Therefore, canonical adaptation was implemented in terms of a tendency for GCs to independently match their forward and reverse sensor levels to the receptor and ligand levels, respectively, on the current target position. (**C**) After implementation of co-adaptation, naïve terminals stop in front of a field of high ephrin-A or EphA ($L_T = 4$, red; $R_T = 4$, blue), respectively, but ignore the same boundary in simulated gap assays with small gap size. In simulations with wider gaps, terminals stop again in front of the second field. Gap size = 20, 40, or 100 units respectively; $n = 12$; $i = 2000$; target field: 200 × 8. (**D**) Co-adaptation predicts that ephrin-A adapted terminals will ignore a field of high EphA after a small gap and *vice versa*. Gap size = 20; other parameters as in (**C**).

The following figure supplement is available for figure 4:

**Figure supplement 1.** Inclusion of co-adaptation does not alter the explanatory power of the computational model.

## Desensitization involves clearance of sensors from the GC surface by clathrin-mediated endocytosis

Searching for the molecular underpinnings of co-adaptation, we transfected RGCs with a SNAP-tagged ephrin-A5 expression construct (pSNAP–ephrin-A5–IRES–EGFP; *Figure 6—figure supplement 1*). The self-labeling SNAP-tag (*Gautier et al., 2008*) in combination with a membrane-

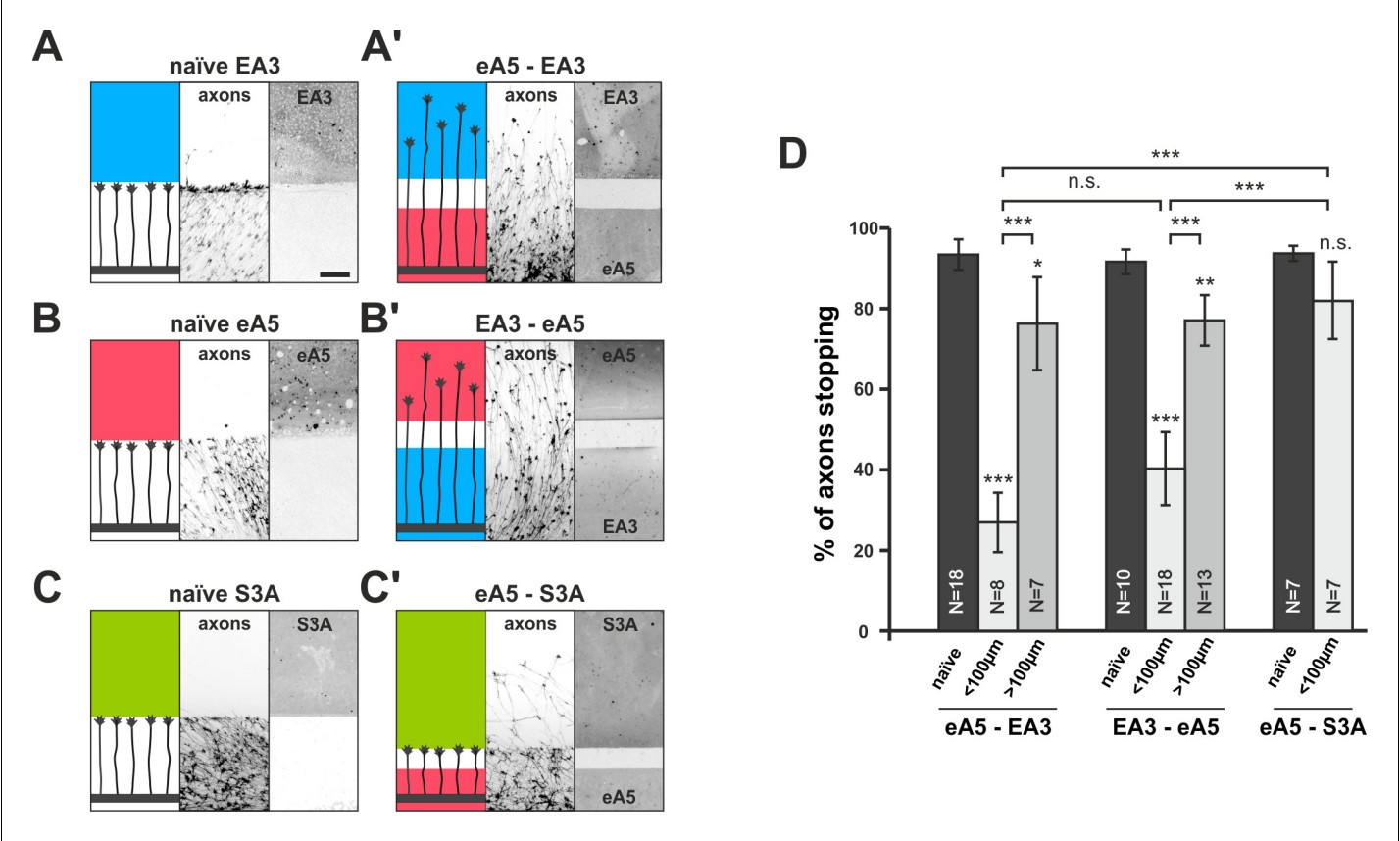

**Figure 5.** Co-adaptation of retinal growth cones in double-cue gap assays. (A–C) Naïve axons stop in front of a homogeneous field of EphA3-Fc (A; EA3, blue; nasal axons), ephrin-A5-Fc (B; eA5, red; temporal axons), or Sema3A-Fc (C; S3A, green; temporal axons). Scale: 100 µm. (A'–C') Nasal, eA5-adapted GCs ignore a field of EA3 after a small gap (A') and, *vice versa*, temporal, EA3-adapted GCs ignore eA5 (B'). By contrast, temporal, eA5-adapted GCs are still strongly repelled by a field of S3A after a small gap, indicating specific co-adaptation of ephrin-A and EphA signals (C'). (D) Quantification of combined data of assays with 65 µm and 90 µm wide gaps grouped in ' < 100 µm' bars; 115 µm and 215 µm grouped in ' > 100 µm' bars. eA5-EA3: naïve: 93.4%, <100µm: 26.9%, >100µm: 76.3% stopping. EA3-eA5: naïve: 91.6%, <100µm: 40.3%, >100µm: 77.1% stopping. eA5-S3A: naïve: 93.7%, <100µm: 81.9% stopping. N: number of independent experiments; error bars indicate standard deviations. T-test with n.s.: $\alpha \geq 0.05$, ***: $\alpha < 0.001$.

The following source data and figure supplements are available for figure 5:

**Source data 1.** Original data underlying bar chart of *Figure 5D*.

**Figure supplement 1.** Growth cone adaptation towards Sema3A.

**Figure supplement 1—source data 1.** Original data underlying bar chart of *Figure 5—figure supplement 1H*.

impermeant fluorescent substrate enabled us to specifically label the surface-bound subpopulation of SNAP–ephrin-A5 in naïve and adapted states. SNAP–ephrin-A5 transfected GCs remain sensitive to substrate-bound ephrin-As (see below). This is in contrast to retrovirally transfected native ephrin-A5, which rendered chick RGC GCs insensitive to external ephrin-As (*Hornberger et al., 1999*). Seemingly, the amount of *cis*-signaling added through the expression of SNAP–ephrin-A5 is sufficiently low to prevent significant disturbance of the guidance system, supporting the validity of the localization probe. On GCs growing on an EphA3-Fc surface, SNAP–ephrin-A5, being the cognate sensor of EphA3, is dramatically reduced compared to control GCs growing on Fc (*Figure 6A,A' and B*). Remarkably, SNAP–ephrin-A5 surface levels are also reduced on GCs growing on ephrin-A5-Fc, despite not being detected by this sensor (*Figure 6A'' and B*). Thus, the reverse signaling sensor is cleared from the surface upon both pure reverse and pure forward signaling, strongly

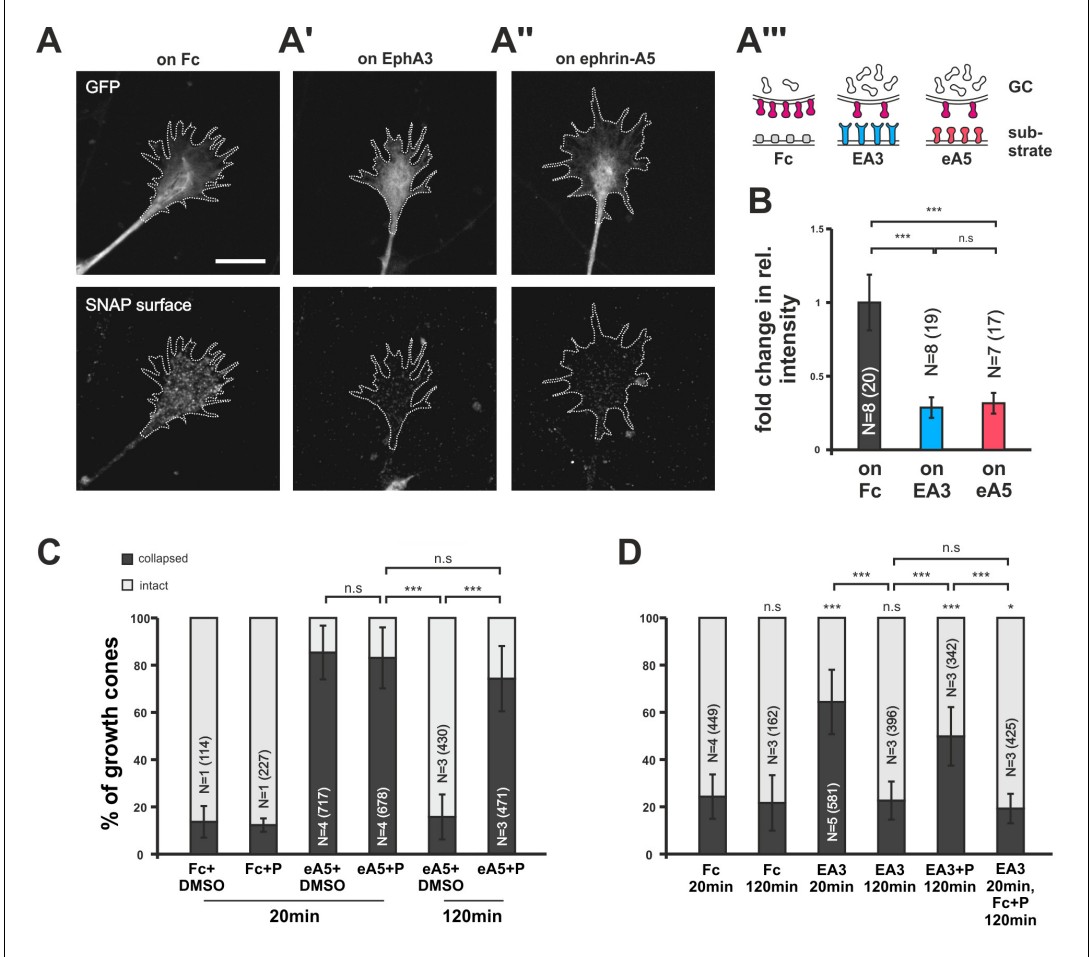

**Figure 6.** Endocytosis of guidance sensors upon growth cone desensitization. (**A**) SNAP–ephrin-A5 surface signal is strongly reduced on GCs growing on either homogeneous EphA3-Fc (**A'**) or ephrin-A5-Fc (**A''**) compared to controls (on Fc), indicating clearance of sensors from the surface upon co-adaptation. Outlines of GCs drawn in the GFP (transfection marker) channel are indicated as white, dotted lines. Scale: 10 μm. (**A'''**) displays a cartoon of the experiment in (**A–A''**) with axonal SNAP surface labeled ephrin-A5 in magenta and with the corresponding guidance cues (Fc, gray; EA3, blue and eA5, red). (**B**) Quantification of SNAP–ephrin-A5 surface signal intensities. Relative signal intensity within the GC outline normalized to the control situation is given as fold change (on Fc: 1; on EA3: 0.29; on eA5: 0.32). (**C**) Effects of Pitstop2 on adaptation towards soluble ephrin-A5. Pitstop2 (P; carrier DMSO) prevents temporal GCs from desensitizing to 0.25 μg/ml ephrin-A5-Fc (eA5+DMSO 120 min: 15.8% collapsed; eA5+P 120 min: 74.3% collapsed). The initial response of GCs towards eA5 is unaltered in the presence of P (eA5+DMSO 20 min: 85.4% collapsed; eA5+P: 83.1% collapsed). Neither P itself, nor its carrier DMSO induces a collapse (Fc+DMSO 20 min: 13.6% collapsed; Fc+P 20 min: 7.5% collapsed). (**D**) Effects of Pitstop2 on adaptation towards soluble EphA3. Soluble EphA3-Fc (EA3; 15 μg/ml) triggers a collapse on nasal or temporal GCs (EA3 20 min: 64.4% collapsed; Fc [4.2 μg/ml] 20 min: 24.3% collapsed). GCs desensitize towards EA3 within 120 min (EA3 120 min: 22.6% collapsed), but not in the presence of 30 μM Pitstop2 (EA3+P 120 min: 49.8% collapsed). P does not impede GCs from recovering their morphology in general (EA3 20 min, then Fc+P 120 min: 19.3% collapsed). Combined data from nasal and temporal GCs. (**B, C, D**) N: number of independent experiments, number of analyzed GCs in brackets; error bars represent standard deviations. T-test with n.s.: α ≥0.05, *: α <0.05, ***: α <0.001.

The following source data and figure supplement are available for figure 6:

**Source data 1.** Original data underlying the bar charts of *Figure 6B—D*.

**Figure supplement 1.** SNAP–ephrin-A5 expression constructs pSNAP–ephrin-A5–IRES-FP contains the coding sequence of a fusion protein consisting of the signal sequence of chick ephrin-A5 (60 bp), the SNAP-tag and full-length chick ephrin-A5 downstream of the CAG enhancer/promoter and upstream of an IRES-FP sequence.

corroborating the co-adaptation concept. By contrast, SNAP–ephrin-A5 surface levels remain unaltered upon adaptation to Sema3A (*Figure 5—figure supplement 1G–H*), again confirming the specificity of the co-adaptation mechanism to the ephrin-A/EphA system.

To address the mechanism of surface clearance during co-adaptation, we used the clathrin-mediated endocytosis (CME) inhibitor Pitstop2 (*von Kleist et al., 2011*). CME is a ubiquitous mechanism for the endocytosis (*McMahon and Boucrot, 2011*) of numerous receptors including EphAs (*Boissier et al., 2013*; *Yoo et al., 2010*). In collapse assays, Pitstop2 does not affect the initial response to the guidance cue (ephrin-A5+DMSO: 85.4%; ephrin-A5+Pitstop2: 83.1% collapsed; *Figure 6C*), nor does it induce a collapse by itself (Fc+DMSO: 13.6%; Fc+Pitstop2: 7.5% collapsed). Strikingly, however, adaptive recovery after prolonged incubation is completely abolished when Pitstop2 is applied together with ephrin-A5 or EphA3 for 120 min (ephrin-A5+DMSO: 15.8%; ephrin-A5+Pitstop2: 74.3%; EphA3: 22.6%; EphA3+Pitstop2: 49.8% collapsed; *Figure 6C and D*), indicating that sensor internalization via CME is required for the desensitization of both channels.

## Re-sensitization involves replenishment of sensors on the GC surface and is independent of local translation

We next asked how desensitized GCs replenish their surface sensor pools upon re-sensitization. Previously described forms of GC adaptation involve local translation for re-sensitization (*Ming et al., 2002*; *Piper et al., 2005*). Thus, we performed ephrin-A5 gap assays in the presence of anisomycin (AIM) to inhibit protein synthesis (*Grollman, 1967*). Consistent with previous reports (*Campbell and Holt, 2001*), AIM treatment did not affect axon elongation until, after about 4 hr, cells start to die (data not shown). We used time-lapse imaging and administered AIM just before axons left the first ephrin-A5 field. Interestingly, AIM-treated GCs display normal re-sensitization, as indicated by the unaltered recognition of the second ephrin-A5-Fc field after a 200 μm wide gap (*Figure 7A*).

As an alternative to local translation, the GC membrane might be repopulated with sensors from internal stores during re-sensitization. To check for this possibility, we specifically labeled the intracellular population of SNAP–ephrin-A5 in desensitized GCs growing on ephrin-A5. Subsequently, GCs were given time (20–22 hr) to leave the ephrin-A5 field for re-sensitization (*Figure 7C*) and were then stained for any formerly intracellular label that had appeared on the GC surface in the meantime (for a detailed staining protocol, see *Figure 7—figure supplement 1* and 'Materials and methods'). Notably, GCs that have left the ephrin field show a significantly higher surface staining than GCs that are still growing on the field (fold change in intensity compared to laminin controls: on ephrin-A5 — 0.85; off ephrin-A5 — 1.13; *Figure 7B–B'' and D*). This indicates a transport of intracellular sensors to the GC's surface upon re-sensitization. Notably, these results were gained for SNAP–ephrin-A5 on ephrin-A5-Fc substrates. As ephrin-A5 is not a receptor for ephrin-A5-Fc, its regulation has to be assumed to be a result of co-regulation with EphAs, the active sensor in this setting, again supporting the co-adaptation idea.

## Co-adaptation implicates temporary storage of guidance sensors in the recycling endosome

Endocytotic clearance and surface repopulation from internal stores upon co-adaptation suggests the involvement of recycling endosomal compartments (*Goldenring, 2015*). To assess whether ephrin-A5 does indeed traffic through recycling endosomes during co-adaptation, we co-transfected RGCs with pSNAP–ephrin-A5–IRES–dTom and pEGFP–Rab11 (*Figure 8A,B,B' and C,C'*). In fact, we find an increased colocalization of SNAP–ephrin-A5 and Rab11 in GCs growing on ephrin-A5 when compared to GCs on laminin (*Figure 8B'', C'' and D*), suggesting elevated partitioning of these sensors in desensitized GCs into Rab11-positive compartments, from which they can return to the surface upon re-sensitization.

## Disrupting membrane rafts induces adaptation

In our model, adaption is adjusted in response to the guidance potential experienced by the GC. A potential regulator of co-adaption should therefore be located downstream of both reverse and forward signaling, which together determine the guidance potential, and it should impact on the membrane activity of the signaling sensors. *Baba et al. (2009)* have shown that Fyn-kinase is activated downstream of ephrin-A reverse signaling, inducing an increase of sphingomyelin in the membrane

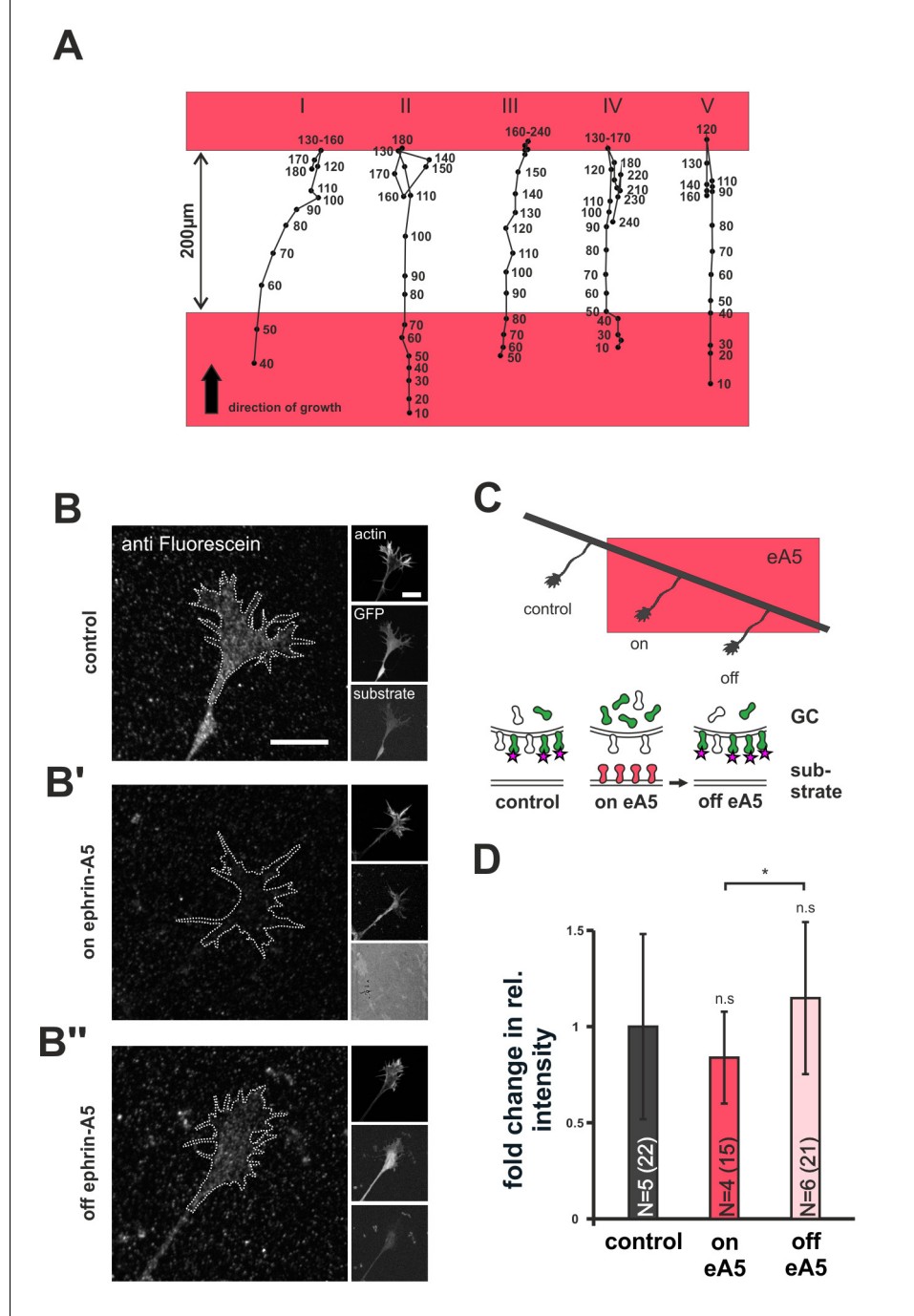

**Figure 7.** Dynamics of SNAP–ephrin-A5 during growth cone re-sensitization. (**A**) Exemplary trajectories of five GCs (combined from N = 3 independent experiments) on ephrin-A5 gap patterns (red) in the presence of 40 μM AIM. Positions of GCs (black dots) were marked at the indicated time points (in minutes) after the addition of AIM. GCs stop and pause (I, III) or stop and retract (II, IV, V) upon reaching the second ephrin field, indicating full re-sensitization despite inhibition of local translation. (**B–B''**) SNAP–ephrin-A5 dynamics upon GC re-sensitization. Representative images of GCs, stained for formerly intracellular SNAP–ephrin-A5 that has been relocated to the membrane. Outlines of GCs drawn in the actin channel are indicated as white, dotted lines. Surface localization of ephrin-A5 (anti-fluorescein signal) is elevated in re-sensitizing GCs (**B''**, off ephrin-A5) compared to desensitized GCs (**B'**, on ephrin-A5). Scale: 10 μm. (**C**) Experimental setup with retinal explant (black) on a field of ephrin-A5 (red) and cartoon of the experiment shown in (**B–B''**) with intracellular SNAP–ephrin-A5 (green), which is detected on the surface (magenta star). (**D**) Quantification of membrane-relocated SNAP–ephrin-A5 signals. Relative anti-fluorescein signal intensity within the GC outline normalized to the control is given as fold change. Control — 1;

*Figure 7 continued on next page*

*Figure 7 continued*

on eA5 — 0.85; off eA5 — 1.13. N: number of independent experiments, number of analyzed GCs in brackets; error bars represent standard deviations. T-test with n.s.: α ≥0.05, *: α <0.05.

The following source data and figure supplement are available for figure 7:

**Source data 1.** Original data underlying the bar chart in *Figure 7D*.

**Figure supplement 1.** Staining procedure for recycled SNAP–ephrin-A5.

and a corresponding exclusion of ephrin-A from the cell surface. Sphingomyelin, together with cholesterol, is a major component of the lipid rafts of the cell membrane (*Simons and Ikonen, 1997*), where ephrin-As have been shown to be localized (*Averaimo et al., 2016*; *Gauthier and Robbins, 2003*). As Fyn-kinase also acts downstream of EphA receptors (*Ellis et al., 1996*; *Wu et al., 1997*), we first asked whether manipulating the sphingomyelin content of the membrane might impact co-adaptation. Pretreatment with sphingomyelinase (SMase, *Neufeld et al., 1996*) in collapse assays strongly reduces the primary response of GCs towards ephrin-A5 and EphA3 (ephrin-A5 — 79.6%; ephrin-A5+SMase — 50%; EphA3 — 52.9%; and EphA3+SMase — 23.8% collapsed; *Figure 9—figure supplement 1A and B*). SMase treatment has been shown to disrupt rafts by replacing them

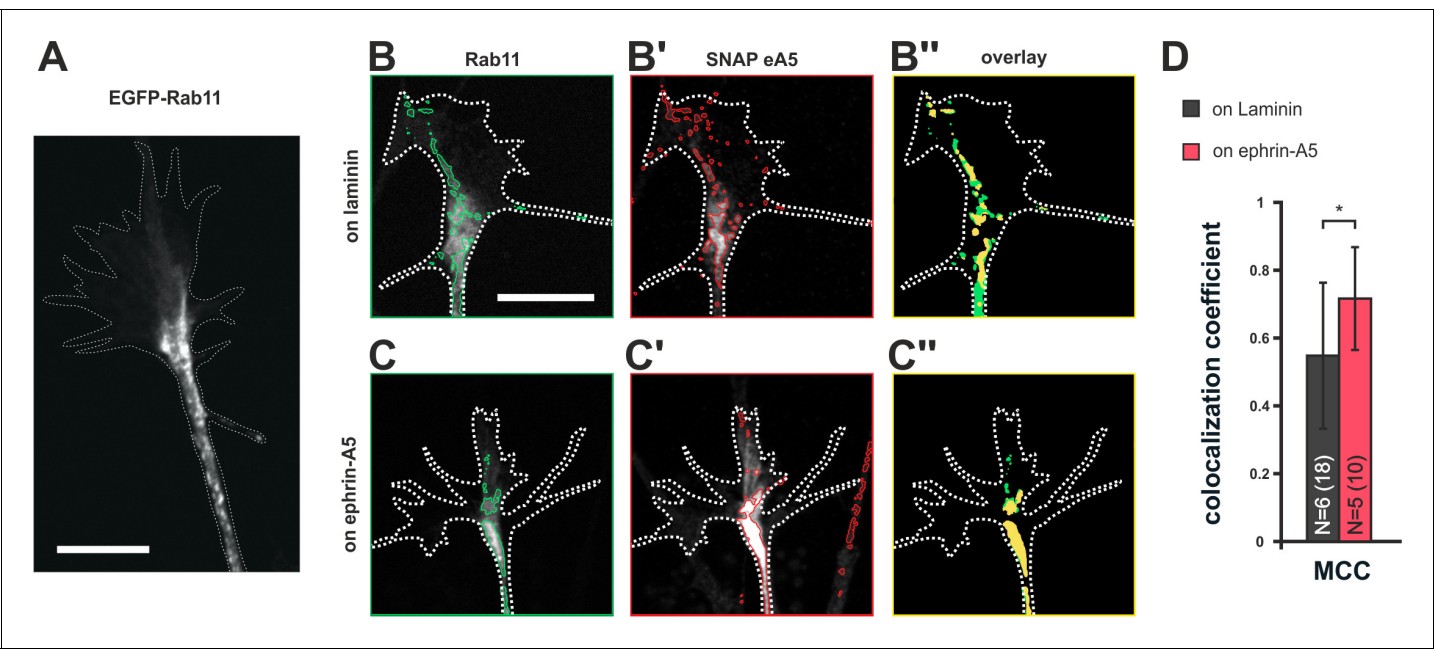

**Figure 8.** Colocalization of SNAP–ephrin-A5 and Rab11-positive endosomes. (**A**) eGFP–Rab11 localization in a transfected RGC axon (outline drawn in the actin channel indicated as white, dotted line). Rab11-positive vesicular structures are predominantly observed in the axon shaft and the central domain of GCs and, to a lesser extent, are found to stretch into the more distal GC. (**B–B′′**) Representative images of an eGFP–Rab11 (**B**) and a SNAP–ephrin-A5 (**B′**) co-transfected GC on laminin substrate. Vesicular structures identified by automated image segmentation are indicated in green and red for each channel, respectively. (**B′′**) Colocalization of SNAP–ephrin-A5 and eGFP–Rab11 within the Rab11-positive vesicles is shown as yellow areas. (**C–C′′**) Example images of an eGFP–Rab11- (**C**) and a SNAP–ephrin-A5-transfected GC (**C′**) growing on ephrin-A5-Fc. (**C′′**) Colocalization of SNAP–ephrin-A5 and eGFP–Rab11 within the Rab11-positive vesicles (yellow areas) increases upon desensitization. (**D**) Manders' colocalization coefficient (MCC) as a measure of colocalization of SNAP–ephrin-A5 and eGFP–Rab11 within the Rab11-positive vesicles in naïve (on laminin: M2 = 0.55) and adapted (on ephrin-A5: M2 = 0.72) GCs, respectively. N: number of independent experiments, number of analyzed GCs in brackets; error bars indicate standard deviations. T-test with *: α <0.05. Scale: 10 µm.

The following source data is available for figure 8:

**Source data 1.** Original data underlying the bar chart of *Figure 8D*.

with ceramide-enriched, densely packed solid ordered domains (*Chiantia et al., 2006*; *Goñi and Alonso, 2009*). Thus, sphingomyelin overproduction or conversion to ceramide might have the same effect of excluding ephrin-As from rafts. To investigate the role of these membrane microdomains in co-adaptation more closely, and to avoid the intricacies of SMase effects, we used methyl-$\beta$-cyclo-dextrin (M$\beta$CD, [*Mahammad and Parmryd, 2015*]) to disrupt rafts by extracting cholesterol from the GC membrane in further adaptation assays. In forward signaling collapse assays with 0.25 µg/ml ephrin-A5-Fc, co-application of 2 mg/ml M$\beta$CD does not significantly alter the primary response after 20 min (80.3% without, 77.4% with M$\beta$CD, *Figure 9A*). After 1 hr of incubation, however, significantly more GCs have recovered in the presence of M$\beta$CD than without (74.2% collapse without, 45% with M$\beta$CD, *Figure 9A*). In reverse signaling collapse assays with 15 µg/ml EphA3, a corresponding desensitizing effect is already seen after 20 min (78.2% collapsed without, 51.6% collapsed with M$\beta$CD, *Figure 9B*) and remains significant after 1 hr (28.3% collapse without, 14.5% with M$\beta$CD, *Figure 9B*). In double-cue gap-assays with a gap size actually allowing for substantial re-sensitization (115 µm), a significantly higher percentage of EphA3- or ephrin-A5-adapted axons are still desensitized when M$\beta$CD is present compared to controls; these axons overgrow the boundary and enter the field of ephrin-A5 or EphA3, respectively, after the gap (EA3-eA5: 80.7% stopping without M$\beta$CD, 46,3% stopping with M$\beta$CD [*Figure 9C,C' and E*]; eA5-EA3: 70.9% stopping without M$\beta$CD, 23.4% stopping with M$\beta$CD [*Figure 9D,D' and E*]). Thus, extraction of cholesterol from the GC membrane, which is thought to result in the disintegration of lipid rafts, strongly enhances desensitization in both types of adaptation assays.

## Modeling suggests that co-adaptation is needed for tectal entry

We have previously shown that our model, through the inclusion of fiber–fiber in addition to fiber–target interactions, displays a substantial degree of mapping plasticity even in the absence of any adaptation (*Weth et al., 2014*). Thus, in accordance with experimental observations, a half-retinal projection expands to a full tectum, a full retinal projection compresses to a half-tectum, and a half-projection can be forced to map onto a mismatching tectal half (*Goodhill and Richards, 1999*). This is due to fiber–fiber interactions, by virtue of which the axons introduce their own mapping apparatus into the target field, eventually overwriting the target cues. Why then should co-adaptation be needed? Generally speaking, signaling comes at the expense of energy. Adaptation is continuously acting to reduce the guidance potential and, thus, the energetic costs of organizing the connectome, which are presumably limiting to the development of the nervous system as the mature brain is the most highly ordered biological structure we know of. A more specific advantage of the adaptation of retinal GCs can be envisaged from corresponding simulations (*Figure 10*). Without adaptation, the majority of axons would be unable to overcome the steep boundary of reverse signaling at the entrance of the target field. By contrast, when we deliberately desensitize the fiber population through the co-adaptation mechanism before entering the target field, they easily overcome the boundary and automatically re-adjust their sensitivities on the target field. To this end, a pre-adapting signal would be required before the target. Fiber–fiber interactions enabled by the intimate contact of the fibers in the optic tract are unlikely to be the preadapting factor, as it has been show that fiber–fiber interactions are required neither for target entry nor for the actual mapping (*Gosse et al., 2008*). This suggests the existence of still elusive biochemical preadapting factors, which should be present in the optic tract, but absent from the target.

## Discussion

The surprising finding of this study is that GCs of chick RGCs can adapt towards forward and reverse ephrin-A/EphA signals, which by their quantities provide the critical cues for topographic guidance. Inspired by computational modeling, we suggest that the seemingly incompatible properties of topographic accuracy and adaptation can be reconciled by a distinctive novel mode of strictly parallelized modulation of the signaling channels, termed co-adaptation. Experimentally, we demonstrate that co-adaptation is implemented in retinal GCs. The cellular mechanism of co-adaptation is based on the trafficking of guidance sensors between the GC membrane and internal stores, including the recycling endosome. Finally, we provide evidence indicating that repartitioning of guidance sensors between membrane microdomains might be a critical first step in co-adaptation. According to

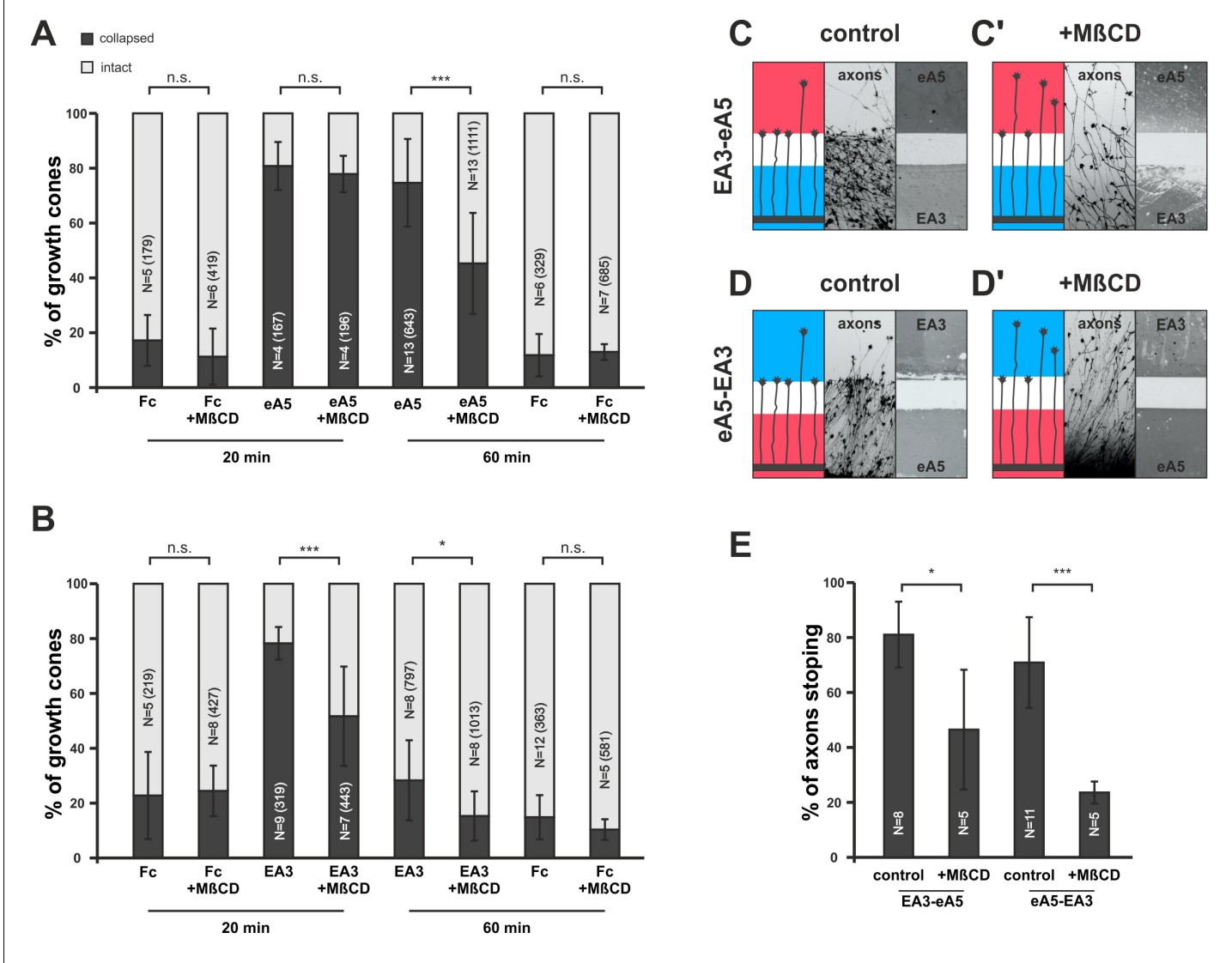

**Figure 9.** Disintegration of lipid microdomains induces co-adaptive growth cone desensitization. (**A**) Ephrin-A5-Fc and (**B**) EphA3-Fc collapse assays. Methyl-beta-cyclodextrin (M$\beta$CD, 2 mg/ml)-treated GCs show enhanced desensitization (reduced collapse rates) when exposed to soluble ephrin-A5-Fc (0.25 µg/ml) or EphA3-Fc (15 µg/ml) compared to controls without M$\beta$CD. (**A**) GCs collapsed after 20 min incubation: Fc (0.25 µg/ml),16.7%; Fc+M$\beta$CD, 11.4%; eA5, 80.3%; and eA5+M$\beta$CD, 77.4%. GCs collapsed after after 60 min incubation: Fc, 12.1%; Fc+M$\beta$CD, 13.4%; eA5, 74.2%; and eA5+M$\beta$CD, 45.0%. (**B**) GCs collapsed after 20 min incubation: Fc (4.2 µg/ml), 22.8%; Fc+M$\beta$CD, 24.4%; EA3, 78.2%; and EA3+M$\beta$CD, 51.6%. GCs collapsed after 60 min incubation: Fc, 14.9%; Fc+M$\beta$CD, 10.4%; EA3, 28.3%; and EA3+M$\beta$CD, 14.5%. N: number of independent experiments, number of analyzed GCs in brackets; error bars show standard deviations. T-test with n.s.: $\alpha \geq 0.05$, *: $\alpha < 0.05$, ***: $\alpha < 0.001$. (**C**) EA3–eA5 and (**D**) eA5–EA3 double-cue gap assays. After 115 µm wide gaps, untreated GCs predominantly stop, whereas M$\beta$CD-treated GCs (**C'**, **D'**) ignore the protein field after the gap. (**E**) Quantification of double-cue gap assays: EA3–eA5, 80.7%; EA3–eA5+M$\beta$CD, 46.3% stopping; eA5–EA3, 70.9%; and eA5–EA3+M$\beta$CD, 23.6% stopping. N: number of independent experiments, error bars show standard deviations, T-test with *: $\alpha < 0.05$, ***: $\alpha < 0.001$.

The following source data and figure supplements are available for figure 9:

**Source data 1.** Original data underlying the bar charts in *Figure 9A, B, E*.

**Figure supplement 1.** Disintegration of lipid microdomains by sphingomyelinase desensitizes growth cones towards soluble ephrin-A5 and EphA3.

**Figure supplement 1—source data 1.** Original data underlying the bar charts of *Figure 9—figure supplement 1*.

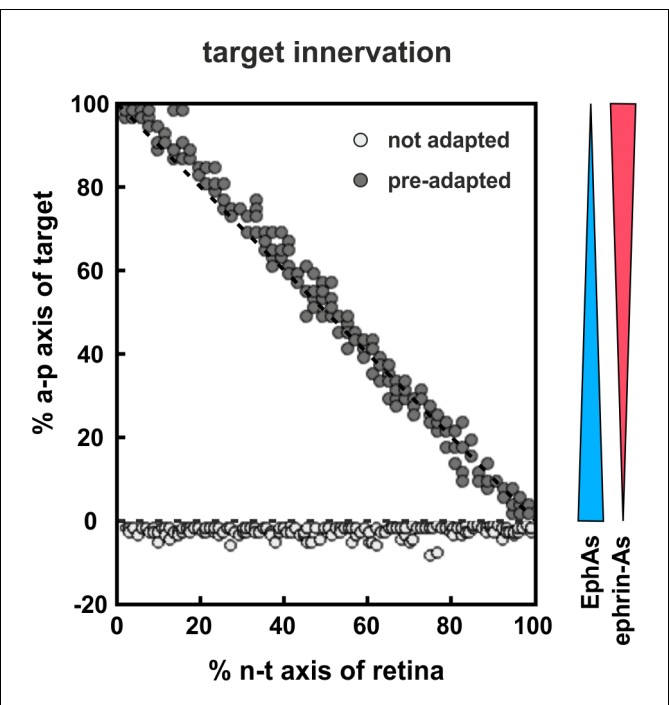

**Figure 10.** Modeling tectal innervation. Co-adaptation enables fiber terminals to enter a tectal target field initially and allows correct mapping therein (gray circles), whereas non-adapted terminals (white circles) fail to enter a tectal target field in simulations with 200 GCs placed in front of the target. The graded distributions of EphAs (blue) and ephrin-As (red) on the target field are symbolized by the colored wedges. $R_F$ and $L_F$ of adapted terminals are initially deflected by a factor of 30.

modeling, co-adaptation could have a particular role during tectal entry, allowing axons to overcome the steep reverse signaling boundary at this location.

## Co-adaptation preserves the ephrin-A/EphA ratio, the genetic imprint of topographic identity

It has repeatedly been shown before that GCs can adapt to axonal guidance cues that act as global attractants or repellents (*Ming et al., 2002*; *Piper et al., 2005*). Adaptation in these cases might be a means of adjusting the dynamic range of the sensors. By contrast, adaptation against topographic cues comes unexpectedly, as these cues have been thought to act by virtue of their concentrations within graded distributions since the first proposition of gradient guidance (*Sperry, 1963*). Concentration gradients provide two kinds of information for chemotactic guidance: positional information, via the local concentration, $c(x)$, and directional information, via the derivative, $dc(x)/dx$. Adaptation amounts to deleting the positional information. This is what is found in bacterial chemotaxis (*Parkinson et al., 2015*). At every location in the gradient, the CheR methyltransferase desensitizes chemotaxis receptors so that upon relocation of the bacterium, only the incremental change of the cue concentration is detected. This exclusive readout of the gradient's directional information enables the bacteria to reach the source or the sink of an attractant or repellent, respectively, but precludes the identification of intermediate targets. The same would hold true for the topographic guidance of GCs.

In our previously published computational model of topographic guidance (*Gebhardt et al., 2012*), the topographic identity of the GCs is encoded in the ratio of ephrin-As to EphAs that they express. This is because this ratio determines the ratio of instantaneous reverse to forward signals received by the GC, which, according to the model, has to be balanced upon target approach. Thus, retaining topographic identity in the face of adaptation requires that these sensing channels are co-modulated concomitantly by the same factor. We challenged this idea of co-adaptation

experimentally by testing the strong prediction that, through co-adaptive coupling, a signaling channel should be modulated even in the absence of its cognate signal, as long as the coupled channel is activated and modulated. We show that this is what happens in retinal GCs with regard to reverse and forward ephrin-A/EphA signaling. The mechanism is specific to the ephrin/Eph system, and GCs remain sensitive to other guidance signals throughout. To the best of our knowledge, co-adaptation is a novel signaling phenomenon not described previously. Canonical adaptation can be conceived as a negative feedback to channel activation. Co-adaptation, by contrast, also modulates coupled, non-activated signaling channels.

It should be noted that the existence of co-adaptation in retinal GCs along the way supports our modeling assumption that topographic identity is encoded in the ephrin-A/EphA ratio. It also corroborates the occasionally disputed role of reverse signaling in topographic mapping. Why should a specific co-adaptation mechanism for the two channels exist, if only one was needed for mapping?

## The cellular mechanism of co-adaptation involves the shuttling of a controlled subpopulation of membrane-bound sensors on a special trafficking route

According to our findings, co-adaptation in retinal GCs is based on vesicular trafficking. Desensitization is correlated with surface clearance of ephrin-As and EphAs by CME, whereas re-sensitization implies replenishment of ephrin-As and EphAs from recycling endosomal compartments. In contrast to the transport of ephrin-As, the trafficking of EphA receptors has been described in detail (*Sabet et al., 2015*). Receptors that have been auto-phosphorylated as a result of ligand binding, as well as receptors that are spontaneously auto-phosphorylated in the absence of a ligand, are internalized into Rab5-positive early endosomes. Ligand binding, as opposed to spontaneous auto-phosphorylation, is thought to induce clustering (*Janes et al., 2012*; *Nikolov et al., 2013*) and additional phosphorylation. The latter recruits ubiquitinylating enzymes, eventually diverting the stable receptor/ligand complexes to the late endosomal/lysosomal compartment for degradation. Unliganded, spontaneously auto-phosphorylated receptors, however, are trafficked to the recycling endosome, bringing them into apposition with tyrosine phosphatase PTP1B on the cytosolic face of the endoplasmic reticulum membrane. After dephosphorylation, the receptors are recycled to the plasma membrane. Despite being triggered by ligand binding, co-adaptation undoubtedly necessitates the trafficking of unbound receptors. It is unlikely, however, that these correspond to the spontaneously auto-phosphorylated subpopulation described above. This is because spontaneous auto-phosphorylation is expected to occur at constant frequency, whereas the degree of co-adaptation must vary with the impinging signal. Thus, we propose that the sensors trafficked during co-adaptation form an independent subpopulation, whose size is adjusted to the required degree of adaptation and which is shuttled on a pathway that is not identical with the routes described above. As co-adaptation requires the shuttling of the proper absolute number of sensors in the topographically appropriate ratio, the question arises as to how this subpopulation might be selected for trafficking.

## Repartitioning between membrane lipid microdomains as the potential mechanism allocating sensors to the co-adaptation pathway

The degree of co-adaptation required in the model is determined by the instantaneous guidance potential, $D$, measured by a GC, *i.e.*, downstream of the integration of reverse and forward signals. Sphingomyelin has been shown to increase in the cell membrane upon activation of Fyn-kinase, downstream of ephrin-A/EphA signaling, and to induce a concomitant decrease of membrane ephrin-As (*Baba et al., 2009*). This sphingomyelin appeared to be a promising candidate regulator of co-adaptation. In fact, SMase pretreatment desensitizes retinal GCs to both ephrin-A forward and EphA reverse signaling.

Sphingomyelin, together with cholesterol, is thought to be a key organizer of lipid rafts, which are liquid-ordered microdomains of the cell membrane (*Róg and Vattulainen, 2014*; *Simons and Sampaio, 2011*). We therefore went on to test the effects of M$\beta$CD, an efficient scavenger of cholesterol, on GC co-adaptation. In fact, we find a strong enhancement of the desensitization of forward and reverse signaling in both collapse and gap co-adaptation assays upon application of M$\beta$CD. Thus, manipulation of both major components of lipid rafts yields the same effect, supporting the idea that these membrane microdomains are involved in the observed

desensitization. Formally, our experiments cannot rule out the possibility that lipid raft disruption simply impedes primary signaling and is thus unrelated to the co-adaptation mechanism. However, the fact that ephrin-A reverse signaling impacts on the relevant lipid composition of the membrane strongly suggests the idea that raft microdomain alteration is a genuine element of the GC co-adaptation mechanism.

Raft disintegration would lead to a release of ephrin-As, which have been shown to be localized in membrane rafts (*Averaimo et al., 2016*; *Gauthier and Robbins, 2003*), into the liquid disordered domain of the membrane. There is substantial evidence that EphAs might also reside in lipid rafts (*Boscher et al., 2012*; *Chakraborty et al., 2012*; *Foster et al., 2003*; *Gu et al., 2012*; *Inder et al., 2012*; *Kim et al., 2012*; *Yang et al., 2010*; *Yi et al., 2013*; *Zheng et al., 2011*), potentially driven by the lipid-interactions of their second fibronectin type III domain (*Chavent et al., 2016*), although these rafts seem to be separate from the ephrin-A-containing microdomains (*Kao and Kania, 2011*; *Marquardt et al., 2005*). Therefore, we suggest that a similar dispersion from microdomains might also happen to EphAs upon variation of the sphingomyelin content of the membrane (*Figure 11*). These processes would eventually generate an intermixed subpopulation of dispersed ephrin-As and EphAs in the non-raft membrane.

Notably, raft-located proteins are typically internalized by clathrin-independent endocytosis (*Sandvig et al., 2011*). By contrast, we observe CME for both ephrin-As and EphAs during co-adaptation, supporting the idea that the endocytosed subpopulation of molecules might not be raft localized. If, as we propose, alteration of the sphingomyelin content induced the same fraction of ephrin-As and EphAs to re-partition from their respective microdomains to the liquid-disordered phase of the membrane, the dispersed mixture eventually would have the same ratio of ephrin-As and EphAs as the whole membrane. It could thus be endocytotically removed from the surface upon co-adaptation without altering the topographically relevant ephrin-A/EphA ratio. The same would be hardly conceivable if membrane clearance involved only ephrin-A/EphA ligand receptor complexes, as their ratio would be stoichiometric instead of depending on the topographic imprint of the respective cell. The disperse subpopulation would probably not be activated, as activated sensors should stay in their respective microdomains due to increased clustering caused by ligand binding (*Janes et al.,*

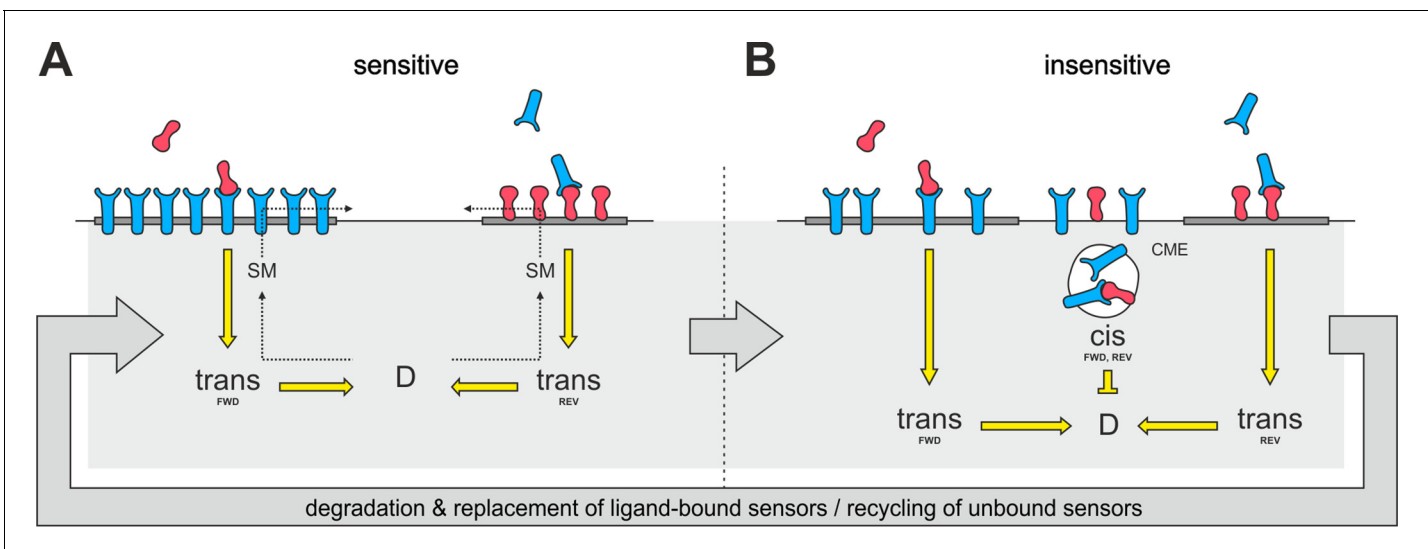

**Figure 11.** Proposed mechanism of co-adaptation. (A) EphAs (blue) and ephrin-As (red), located in membrane lipid microdomains (dark gray), signal in *trans* due to external cues. *Trans* forward (FWD) and reverse (REV) signals integrate into the guidance potential, *D*, and elicit repulsion when not balanced. *Trans* signaling also promotes sphingomyelin (SM) synthesis, and when incorporated into the membrane, this phospholipid might proportionally solubilize EphAs and ephrin-As from their respective domains. (B) Dispersed, unbound sensors are destined for constitutive clathrin-mediated endocytosis (CME) and might increase *cis* signaling upon internalization, as anti-parallel orientation is favored in high-curvature membrane vesicles. Enhanced *cis* signals desensitize GCs to *trans* signals. Sensitivity returns with the recycling of unbound sensors, and degraded receptor/ligand complexes should constitutively be replaced through protein synthesis in the long run.

*2012*; *Nikolov et al., 2013*). Thus, the co-adaptive subpopulation would not be expected to carry any phosphorylation tag to trigger endocytosis. We instead suggest that the disperse distribution by itself might destine for constitutive endocytosis (*Figure 11*).

### Increased *cis*-signaling from endosomes as the potential core mechanism of co-adaptation

Albeit highly intuitive that desensitization and re-sensitization might involve sensor endocytosis and recycling, respectively, our computational modeling suggests unexpected twists on the interpretation of sensor trafficking. Surprisingly, according to the model, the sensor surface concentration is irrelevant to adaptation. This can be seen, for example, when an individual axon is growing on a homogeneous ephrin target. Then, the model equation reduces to

$$D = |ln\frac{1}{L_T/L_F + 1}|$$

which includes the ephrin on the fiber terminal, $L_F$, and on the target, $L_T$, but not on the receptor actually detecting the *trans* signal. In fact, the model suggests that the axonal ligand representing *cis*-signaling is the relevant parameter determining the guidance potential. As discussed in the previous paper (*Gebhardt et al., 2012*), we assume *cis*-interactions to be signal transducing instead of signal inactivating. That it is, in fact, *cis*-signaling that governs co-adaptation in the model can be seen more easily when the model equation for an individual axon is expressed conceptually in terms of signaling channels (REV: reverse, FWD: forward):

$$D = |ln\frac{REV_{trans} + REV_{cis}}{FWD_{trans} + FWD_{cis}}|$$

Upon introduction of an adaptation factor, *a*, which was originally meant to equally modulate all sensor concentrations, the model equation becomes

$$D = |ln\frac{a*REV_{trans} + a^2*REV_{cis}}{a*FWD_{trans} + a^2*FWD_{cis}}|$$

The square in front of the *cis*-signaling is due to the fact that, in contrast to *trans*-signaling, *cis*-signaling involves both axonal sensors. This obviously reduces to

$$D = |ln\frac{REV_{trans} + a*REV_{cis}}{FWD_{trans} + a*FWD_{cis}}|$$

Thus, effectively, only the efficiency of *cis*-signaling is modulated during co-adaptation. Figuratively, this compares to a situation in which we listen to music via earphones (*cis*-signaling) to suppress external acoustic signals (*trans*-signaling).

The proposed type of signal transducing *cis*-interactions has the steric requirement that receptor and ligand be aligned, as in trans-signaling, in an anti-parallel orientation (*Himanen et al., 2001*). This will normally occur only occasionally, when filopodia collide or when membrane ruffles are formed. Thus, the major effect of endocytosis in co-adaptation (*Figure 11*) might be to provide high membrane curvature (around the vesicle lumen) to increase the efficiency of *cis*-signaling from endosomes (*Schmick and Bastiaens, 2014*).

In summary, the proposed mechanisms could explain how co-adaptation might reconcile targeting accuracy in topographically mapping GCs with their adaptability. Thus, our study reveals that the primary task of forming a proper topographic map, despite being based on genetic hardwiring, is achieved through an astonishingly flexible mechanism.

## Material and methods

Reagents were obtained from Thermo Fisher Scientific, Waltham, MA, USA unless explicitly stated.

### RGC explant cultures

Retinae of embryonic day 6 to 7 (E6–E7) chick embryos were dissected in ice-cold Hanks' balanced salt solution (HBSS), sucked onto a nitrocellulose filter and cut into 250–300 µm wide naso-temporal

strips. Explants were placed on cover slips coated with 20 µg/ml mouse natural laminin in HBSS (1 hr at 37°C) and grown in F12 medium containing 0.4% methylcellulose, 5% fetal calf and 2% chicken serum (F12-MC) at 37°C and 4% $CO_2$ for 20–24 hr.

## Collapse assays

The medium on explant cultures was replaced with warm F12-MC containing recombinant human ephrin-A5-Fc, mouse EphA3-Fc (both RnD Systems, Minneapolis, MN, USA) or human Fc fragment (Calbiochem, San Diego, CA, USA). For inhibitor experiments, 30 µM Pitstop2 (Abcam, Cambridge, UK) or 400 mU/ml sphingomyelinase (Sigma-Aldrich, St. Louis, MO, USA) in F12-MC was applied during a 15 or 30 min pre-incubation, respectively, and thereafter together with ephrin-A5-Fc, EphA3-Fc or Fc. Methyl-$\beta$-cyclodextrin (2 mg/ml; Sigma-Aldrich, St. Louis, MO, USA) was administered together with ephrin-A5-Fc, EphA3-Fc or Fc. After incubation for 20 or 120 min, cultures were fixed and stained with Alexa488 or Alexa568-phalloidin and the percentage of collapsed GCs was counted.

## Gap assays

Gap patterns were produced by direct contact printing as described previously (*von Philipsborn et al., 2006a*). In short, a PDMS stamp (Elastosil RT 625; Wacker Chemie, Munich, Germany) comprising the gap pattern was covered with protein solution for 2 hr at 37°C, rinsed in $H_2O$, dried under a stream of nitrogen gas and then stamped onto an epoxysilanized and poly-L-lysine (PLL)-covered glass coverslip (Ø 18 mm, VWR, Radnor, PA, USA) for 15 min at 37°C (cleaned coverslips in 1% [3-glycidoxypropyl]-trimethoxysilane [abcr, Karlsruhe, Germany] in pure EtOH, pH 5.5 for 5 min, air dried and covered with 200 µg/ml PLL [Sigma-Aldrich, St. Louis, MO, USA] in PBS overnight). The substrate was then covered with laminin, rinsed in $H_2O$, covered with F12 and kept at 37°C until imminent usage.

For single-cue gap assays 15 µg/ml ephrin-A5-Fc, 15 µg/ml EphA3-Fc or 25 µg/ml mouse semaphorin3A-Fc in PBS were used. For double-cue gap assays, both fields of the stamp were physically separated using a piece of Parafilm™ and each was covered with a different protein solution (combinations of 15 µg/ml ephrin-A5-Fc, 15 µg/ml EphA3-Fc and 25 µg/ml mouse semaphorin3A-Fc in PBS). The pattern orientation was indicated on the coverslip. Patterns were stained with Alexa-conjugated anti-human goat IgGs. Quantification of gap assays was performed using a custom-written MATLAB (RRID:SCR_001622) code (*Weth, 2017*) to count fibers in and immediately behind the gap. In short, grayscale images of axons were thresholded and two ROIs were drawn in parallel to the gap (ROI one in the gap, ROI two directly behind the gap; both ~20 µm wide and over the complete lateral extension of the gap). Both ROIs were then evaluated, counting signal peaks in each pixel row, and the percentage of stopping fibers was calculated from mean counts in ROI one and ROI 2. Methyl-$\beta$-cyclodextrin (2 mg/ml; Sigma-Aldrich, St. Louis, MO, USA) was directly added to the culture medium. For experiments with anisomycin (AIM, Sigma-Aldrich, St. Louis, MO, USA), 40 µM AIM was added to the culture medium when axons were just about to leave the first ephrin-A5 field, and GCs were tracked using time-lapse imaging.

## Expression constructs

To construct pSNAP–ephrin-A5–IRES–EGFP (*Figure 6—figure supplement 1*), we first reverse transcribed and PCR amplified the message for full-length *Gallus gallus* ephrin-A5 (NCBI [RRID:SCR_006472] NM_205184.2) using primers starting 16 bp upstream of the start and ending 27 bp downstream of the stop codon from total RNA of the E7 chick tectum. Using SdaI and NotI linkers, the amplification product was inserted between the respective sites of the pSNAP-tag(m) vector (New England Biolabs, Ipswich, MA, USA). In addition, a synthetic double-stranded DNA encoding the first 20 amino acids of chick ephrin-A5 (signal peptide) was inserted into the EcoRV site of the same construct to create a continuous open reading frame for a fusion protein containing the ephrin-A5 signal peptide, the SNAP-tag and full-length ephrin-A5. This coding sequence was amplified from the construct using the primers starting 8 bp upstream of the start codon and ending 36 bp downstream of the stop codon, and was inserted into the SmaI site of the expression vector pCIG2, containing a CAG enhancer/promotor upstream and IRES–EGFP downstream of the insert (*Hand et al., 2005*). For pSNAP-ephrin-A5-IRES-dTom, the EGFP coding sequence was excised using MscI from pSNAP-

ephrin-A5-IRES-EGFP and replaced by the coding sequence for dTomato amplified from pCAGJC-dTomato (*Lee et al., 2013*) using MscI primer linkers starting 4 bp upstream and ending 33 bp downstream of the dTomato coding sequence. pEGFP-Rab11 was a kind gift of Dr. Esther Stoeckli (University of Zurich). It contains the coding sequence of human Rab11 N-terminally fused to EGFP using the pEGFP-C3 vector (TaKaRa Bio USA, Mountain View, CA, USA), which contains the CMV enhancer/promotor (*Alther et al., 2016*).

## Electroporation

For transgenic SNAP-ephrin-A5 expression, dissected retinae were cut and treated with 100 µl ice-cold Accutase solution for 5 min at room temperature and electroporated using 330 ng/µl pSNAP-ephrin-A5-IRES-EGFP/dTomato plasmid in 0.5x PBS and CUY700P20 electrodes (NepaGene, Ichi-kawa-City, Japan; 15V, 5 × 50 ms, 950 ms off time). Because of higher transfection efficiency, we later switched to whole-eye electroporation as described in *Vergara et al., 2013* using 1 µg/µl pSNAP-ephrin-A5-IRES-EGFP/dTomato and 1 µg/µl pEGFP–Rab11.

## SNAP labeling

All SNAP labeling reagents were purchased from New England Biolabs, Ipswich, MA, USA and administered at 1 µg/ml in warm F12-MC for 40 min at 37°C. For Surface SNAP-ephrin-A5 staining, SNAP Surface488 reagent was followed by washing with warm F12-MC. After fixation, cells were treated with sodium borohydride solution (2 × 5 min, about 250 mM in PBS) to remove background and stained with anti-AlexaFluor488 rabbit IgG (RRID:AB_221544) and anti-rabbit AlexaFluor647 goat IgG (Jackson ImmunoResearch, Suffolk, UK) to improve signal/noise. For recycling assays (*Figure 7—figure supplement 1*) surface SNAP-ephrin-A5 was blocked using SNAP Surface Block. Intracellular SNAP-ephrin-A5 was labeled with SNAP Cell Fluorescein followed by washing with warm F12-MC. Cultures were grown for another 20–22 hr before application of anti-fluorescein mouse IgG in pre-warmed F12-MC to the living cells for 15 min. After washing, cells were fixed and stained with anti-mouse AlexaFluor647 goat IgG for formerly intracellular SNAP-ephrin-A5 now relocated to the cell surface. Images were taken using the ApoTome module on a Zeiss AxioimagerZ1 microscope (Zeiss, Oberkochen, Germany).

## Colocalization

Colocalization of SNAP–ephrin-A5 with EGFP–Rab11 was evaluated by calculating the Manders' colocalization coefficients M1 and M2 using the Fiji (RRID:SCR_002285) plugin Coloc2 (https://github.com/fiji/Colocalisation_Analysis/, V2.0.2) after image segmentation to isolate vesicular structures within GCs. Segmentation was performed using Squassh (*Rizk et al., 2014*) in Fiji with parameters: regul.=0.05; min.obj.int.=0.09; subpix.seg=true; excl.z-edge=true; localint.est.=auto; noise=gauss; PSFstd.dev.xy=0.7, z = 0.8; removeint.=0; and removesize = 2.

## Simulations

All simulations were performed using MATLAB 8.4 (RRID:SCR_001622, The MathWorks, Natrick, MA, USA). Except for the introduction of co-adaptation (see below), all modeling conditions were as previously described (*Gebhardt et al., 2012*). Briefly, the projection target, $T$, a rectangular array of unit squares ($X_{T,max}*X_{T,max} = 50*8$), displays the guidance cues along the x-axis (exponential counter-distributions of ephrin-As [$L_T$] and EphAs [$R_T$] for simulations of tectal innervation and mapping, step-functions for simulation of gap assays). Fiber terminals, $F$, are modeled as circular discs with a diameter of about seven units, carrying ligands ($L_F$ for terminal under consideration, $L_f$, for interacting terminals), and receptors ($R_F$ and $R_f$, respectively) in Gaussian-shaped distributions, according to their origin in the retina, which also bears exponential counter-distributions of $L_F$ and $R_F$, and from which the terminals are equally sampled. On the target field, terminals are allowed to overlap freely while performing a random walk, which is biased by the tendency to minimize a guidance potential, $D$. In every iteration, $i$, with current terminal center position $(x_T^*, y_T^*)$, $D_{F,i}(x_T^*, y_T^*)$ is calculated from total EphA forward and total ephrin-A reverse signals, both comprising fiber–target, fiber–fiber and *cis* interactions (assumed to be signal transducing in both directions). All signals are calculated from mass action for the corresponding receptors and ligands over all unit increments of the target and of other terminals overlapped by the terminal under consideration. All interactions are weighted

equally (with constants set to one), except *trans* fiber signals, whose influence, $C(i)$, increases with iteration number to conceptually reflect the developmental increase in terminal number and size. Thus,

$$D_{F,i}\left(x_T^*, y_T^*\right) = \left| ln\left( \frac{\sum_{x_T, y_T} L_F(x_T, y_T)\left[R_T(x_T, y_T) + R_F(x_T, y_T) + C(i)\, R_f(x_T, y_T)\right]}{\sum_{x_T, y_T} R_F(x_T, y_T)\left[L_T(x_T, y_T) + L_F(x_T, y_T) + C(i)\, L_f(x_T, y_T)\right]} \right) \right| \tag{1}$$

with the nominator representing total reverse and the denominator total forward signals instantaneously impinging on the terminal. A GC has reached its target position, when both signaling channels are balanced (total reverse/total forward = 1) and, therefore, when $D$ is minimized $(abs(ln(1)) = 0)$. For more detail, see *Gebhardt et al. (2012)*.

We updated this model to include co-adaptation by introducing a common adaptation coefficient, $a(i)$, and a Hookian resetting force, $f(i)$, modulating both ligands and receptors (collectively called sensors, *S*) on the fiber terminals at every iteration, $i + 1$, depending on the recent history ($h$) of *D*:

$$S(i+1) = a(i)\, S(i) + f(i) \tag{2}$$

with

$$a(i) = 1 + ln\left( 1 + \mu\left( \frac{\sum_{k=1}^{h} k(D(i-h+k))}{\sum_{k=1}^{h} k} \right) \right) \tag{3}$$

and

$$f(i) = \lambda\,(S(0) - S(i)) \tag{4}$$

$\mu$ and $\lambda$ are adjusting parameters. Unless explicitly stated, a unique set of parameters was used in all simulations: number of terminals $n$ = 200; iterations $i$ = 30000; μ = 0.006; λ = 0.0045; and $h$ = 10. For the simulation of *in-vitro* assays (*Figure 4*), where the area density of terminals is sparse, and for the analysis of tectal entry (*Figure 10*), where a role of fiber–fiber interactions has been excluded experimentally, the disproportionate increase of the weight of fiber–fiber interactions was switched off ($C_0$ = 1). On target fields without guidance cues (gap assays and in front of the tectum in the analysis of tectal entry), the random walk of the terminals was given a slight inherent bias to avoid prolonged dwelling ($q_x$ = 0.37). For details on $C_0$ and $q_x$ and further parameters of the basic model, see *Gebhardt et al. (2012)*. Code is available via GitHub (*Weth, 2017*). A copy is archived at https://github.com/elifesciences-publications/RTP_Co-adapt_Model.

## Measurement of ephrin-A5/EphA3 dissociation constants

$K_D$s were measured using the BLItz biolayer interferometer (Pall ForteBio, Menlo Park, USA) at different EphA concentrations (cf. *Figure 2—figure supplement 1*). Streptavidin-coated biosensors were loaded with 5 µg/ml biotinylated human ephrin-A5-Fc (RnD Systems, Minneapolis, MN, USA) and the binding kinetics were measured using concentration series of 0, 3, 15, 33 and 100 µg/ml EphA3-Fc all in BLItz kinetics buffer (loading — 120 s; baseline — 30 s; association — 120 s; and dissociation —120 s). $K_D$s were calculated from on and off rates derived by software from local curve fits corrected for start of association and dissociation.

## Acknowledgements

We would like to thank Dr. Juergen Loeschinger and Dr. Friedrich Bonhoeffer for their support in the early phases of the project, Dr. Christoph Gebhardt for initial contributions and ongoing discussions on the modeling of GC adaptation, Gabi Gerdon for help with the collapse assays, and Dr. Esther Stoeckli for kindly providing the pEGFP–Rab11 plasmid. We are particularly grateful to the reviewers for their thoughtful comments, which substantially improved the paper.

## Additional information

### Funding

| Funder | Grant reference number | Author |
| --- | --- | --- |
| Karlsruhe School of Optics and Photonics | DFG GSC 21 | Felix Fiederling |
| Baden-Württemberg Stiftung | State Scholarship | Felix Fiederling |
| Deutsche Forschungsgemeinschaft | BA 1034/14-3 | Markus Weschenfelder<br>Martin Fritz<br>Martin Bastmeyer<br>Franco Weth |
| Studienstiftung des Deutschen Volkes | PhD Scholarship | Anne  von Philipsborn |
| Deutsche Forschungsgemeinschaft | Open Access Publishing Fund of KIT | Franco Weth |

The funders had no role in study design, data collection and interpretation, or the decision to submit the work for publication.

### Author contributions

FF, Conceptualization, Data curation, Software, Formal analysis, Validation, Investigation, Visualization, Methodology, Writing—original draft, Writing—review and editing; MW, AvP, Data curation, Formal analysis, Validation, Investigation, Visualization, Methodology; MF, Data curation, Formal analysis, Investigation, Visualization; MB, Resources, Supervision, Funding acquisition; FW, Conceptualization, Formal analysis, Supervision, Funding acquisition, Visualization, Writing—original draft, Writing—review and editing

### Author ORCIDs

Felix Fiederling, http://orcid.org/0000-0001-7837-5556

Anne von Philipsborn, http://orcid.org/0000-0002-7921-8744

Franco Weth, http://orcid.org/0000-0002-6819-7028

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
