## [Decision Letter]

Thank you for submitting your article "Ephrin-A/EphA specific co-adaptation as a novel mechanism in topographic axon guidance" for consideration by *eLife*. Your article has been reviewed by two peer reviewers and a Reviewing Editor, and the evaluation has been overseen by the Reviewing Editor and Marianne Bronner as the Senior Editor. The reviewers have opted to remain anonymous.

The Reviewers and Reviewing Editor have discussed the reviews with one another and the Reviewing Editor has drafted this decision to help you prepare a revised submission.

Summary:

Fiederlin and co-workers address a fundamental problem in topographic map formation; the ability of axons to adapt to external guidance cues (de/re-sensitisation) on the one hand, and the requirement of growth cones for quantitative signaling to enable them to orient within a guidance gradient on the other hand. The present study using in vitro signaling assays and computational modeling in an attempt to reconcile this conundrum and advance the field.

The authors have worked for some time on a (molecular) understanding of topographic mapping in the chick retinotectal projection in vitro, in particular focussing on the EphA/ephrin-A family. The new and exciting finding of the present study is the identification and characterization of co-adaptation between forward and reverse ephrin-A/EphA signaling modules; Adaptation in forward signaling is accompanied by adaptation in reverse signaling, thereby maintaining topographically important information and a balance of forward and reverse signaling. The authors' findings are supported and complemented by computational modeling approaches, which in turn provide further insights into potential mechanisms of co-adaptation. Overall, the experiments are of high quality and well documented, and the study is of general interest because it provides a clearer understanding of the molecular mechanisms of topographic map formation.

Essential revisions:

1) The modeling predictions of disrupting co-adaptation are intriguing but remain untested experimentally. We would like to see additional in vitro manipulations showing that co-adaptation is disrupted following treatment with sphingomyelinase. One possible experiment to address this would be the Figure 5 co-adaptation assay repeated in the presence of sphingomyelinase.

2) The interplay (or lack thereof) between the Eph/ephrin system and Sema3A is an interesting feature of the system and should be expanded to elevate the scope and impact of the work. What is the behavior of retinal axons in a gap assay with Sema3A only? What is the outcome of a gap assay in which a Sema3A source is followed – after the gap – by an ephrin-A5 substrate? Does Sema3A alter ephrinA or EphA surface localization?

---

## [Author Response]

*Essential revisions:*

*1) The modeling predictions of disrupting co-adaptation are intriguing but remain untested experimentally. We would like to see additional* in vitro *manipulations showing that co-adaptation is disrupted following treatment with sphingomyelinase. One possible experiment to address this would be the Figure 5 co-adaptation assay repeated in the presence of sphingomyelinase.*

To respond to this point, we have included a complete new set of experiments. The results are presented in the new Figure 9, explained in the Results paragraph “Disrupting membrane rafts induces adaptation” and discussed further in the Discussion paragraph “Repartitioning between membrane lipid microdomains as a potential mechanism of allocating sensors to the co-adaptation pathway”. The old Figure 9 has been moved to the supplement (now Figure 9—figure supplement 1). Corresponding additions have been made to the Materials and methods paragraphs “Collapse assays” and “Gap assays”.

Briefly, we turned from sphingomyelinase to methyl-β-cyclodextrin (MβCD) to manipulate lipid rafts and to study their influence on co-adaptation. SMase generates ceramide, which by itself forms new lipid microdomains with cholesterol (Chiantia et al., 2006; Goni and Alonso, 2009), potentially confounding the interpretation of the results. In addition, the activity of SMase in overnight culture media is hard to control. MβCD, in contrast, is a well-established drug to disrupt lipid rafts by cholesterol scavenging (Mahammad and Parmryd, 2015). We applied MβCD in collapse as well as in co-adaption gap assays. In both cases, we find a profound desensitizing effect of the drug, confirming the previous SMase data. In both forward and reverse signaling collapse assays, axons regenerate faster in the presence of the drug, corresponding to more efficient desensitization. In ephrin-A5 – EphA3 as well as in EphA3 – ephrin-A5 co-adaptation gap assays, axons also are more desensitized, indicated by overgrowing larger gaps. These results strongly support the proposed mechanism of co-adaptation (Figure 11), predicting that raft disintegration would release ephrin-As and EphAs into the liquid-disordered phase of the membrane, from where they are constitutively endocytosed to desensitize the growth cone.

*2) The interplay (or lack thereof) between the Eph/ephrin system and Sema3A is an interesting feature of the system and should be expanded to elevate the scope and impact of the work. What is the behavior of retinal axons in a gap assay with Sema3A only? What is the outcome of a gap assay in which a Sema3A source is followed – after the gap – by an ephrin-A5 substrate? Does Sema3A alter ephrinA or EphA surface localization?*

To answer these questions, we have included another new set of experiments. The results are shown in the new Figure 5—figure supplement 1. They are eluded to in the Results paragraphs “EphA forward signaling in fact co-adapts ephrin-A reverse signaling and vice versa” (second paragraph) and “Desensitization involves clearance of sensors from the GC surface by clathrin-mediated endocytosis” (first paragraph). Necessary additions have been made in the Materials and methods paragraph “Gap assays”.

Briefly, retinal axons display very efficient desensitization towards Sema3A in gap-assays. The desensitization is so efficient that we do not see obvious re-sensitization even with the largest gaps (215µm) available with our contact-printing stamps. Along the way, we observed an interesting behavior of the RGC axons on the Sema3A gap substrates: As soon as the adapted axons leave the Sema3A pedestal and continue to grow on laminin in the gap, they display a strong tendency to defasciculate and to form branches. Dwelling deeper into this phenomenon is, however, beyond the scope of this manuscript.

In Sema3A – ephrin-A5 double-cue gap assays, we observe no co-adaptation. Correspondingly, the SNAP-ephrin-A5 surface localization probe is not internalized on Sema3A. These observations clearly indicate that the two guidance systems (ephrin-A/EphA and Sema3A) do not interplay in retinal growth cones and display an orthogonal behavior. This finding supports the idea, that co-adaptation is a specific feature of the topographic guidance system.